# Direct tissue-sensing reprograms TLR4+ Tfh-like cells inflammatory profile in the joints of rheumatoid arthritis patients

Daniela Amaral-Silva [1,2], Rute Gonçalves[1,2], Rita C. Torrão[1,2], Rita Torres[2,3,4], Sandra Falcão[2,3,4], Maria João Gonçalves[3], Maria Paula Araújo[3], Maria José Martins[3], Carina Lopes[3], Agna Neto[2,3,4], José Marona[3], Tiago Costa[3], Walter Castelão[3], Ana Bento Silva [3], Inês Silva[3], Maria Helena Lourenço[3], Margarida Mateus[3], Nuno Pina Gonçalves[2,3,4], Santiago Manica[2,3,4], Manuela Costa[3], Fernando M. Pimentel-Santos[2,3,4], Ana Filipa Mourão[2,3,4], Jaime C. Branco[3,4,5] & Helena Soares [1,2,6] ✉

CD4+ T cells mediate rheumatoid arthritis (RA) pathogenesis through both antibody-dependent and independent mechanisms. It remains unclear how synovial microenvironment impinges on CD4+ T cells pathogenic functions. Here, we identified a TLR4+ follicular helper T (Tfh) cell-like population present in the blood and expanded in synovial fluid. TLR4+ T cells possess a two-pronged pathogenic activity whereby direct TLR4+ engagement by endogenous ligands in the arthritic joint reprograms them from an IL-21 response, known to sponsor antibody production towards an IL-17 inflammatory program recognized to fuel tissue damage. Ex vivo, synovial fluid TLR4+ T cells produced IL-17, but not IL-21. Blocking TLR4 signaling with a specific inhibitor impaired IL-17 production in response to synovial fluid recognition. Mechanistically, we unveiled that T-cell HLA-DR regulates their TLR4 expression. TLR4+ T cells appear to uniquely reconcile an ability to promote systemic antibody production with a local synovial driven tissue damage program.

[1] Human Immunobiology and Pathogenesis Group, Lisboa, Portugal. [2] iNOVA4Health | CEDOC, NOVA Medical School | Faculdade de Ciências Médicas, NOVA University of Lisbon, Lisboa, Portugal. [3] Hospital Egas Moniz, Rua da Junqueira nº 126, Lisboa, Portugal. [4] Rheumatological Diseases Laboratory, Lisboa, Portugal. [5] CHRC|CEDOC, NOVA Medical School | Faculdade de Ciências Médicas, NOVA University of Lisbon, Lisboa, Portugal. [6] The Discoveries Centre for Regenerative and Precision Medicine, Lisbon Campus, Rua do Instituto Bacteriológico 5, Lisboa, Portugal. ✉email: helena.soares@nms.unl.pt

In rheumatoid arthritis (RA) combined immune and joint tissue dysregulation synergize in propagating chronic inflammation and articular destruction. CD4$^+$ T cells have been strongly implicated in RA pathogenesis through both antibody-dependent and independent mechanisms[1,2]. It remains unclear, however, which CD4$^+$ T-cell population drives RA and how the joint microenvironment impinges on their pathogenic functions. Unveiling CD4$^+$ T-cell pathogenic phenotype and its crosstalk with the arthritic joint environment would benefit diagnosis, patient stratification and could contribute to the design of better drugs that could effectively induce remission.

Effector functions sponsored by CD4 T cells in the joints constitute an active field of research. Circulating CXCR5$^+$ and/or PD1$^+$CXCR5$^+$ Tfh cell populations have been correlated with B cell expansion and increased disease activity[3–5]. Notwithstanding, CD4$^+$ T-cell-mediated antibody-independent mechanisms are at play in RA pathogenesis. Namely, IL-17 production by CD4$^+$ T cells has been implicated in bone erosions[6,7] and cartilage damage[8–10], with its neutralization reducing disease activity[11] and curtailing cartilage and bone damage[6]. IL-17 production is regulated locally at the affected joint[12], requiring both propitious tissue environment and cell-cell interactions, making it challenging to characterize IL-17 producing CD4$^+$ T cells in RA.

T-cell effector programs are profoundly shaped by the local tissue microenvironments where antigen recognition occurs[13]. RA joints are enriched in endogenous pro-inflammatory molecules and in pathogen recognition receptors that recognize them, namely Toll-Like Receptors (TLRs). Polymorphisms in TLR4 have been associated with increased RA susceptibility in humans[14] and mice with TLR4 impairments are protected from experimental arthritis[15–17]. In addition, TLR4 and its endogenous ligands are elevated in the synovial fluid and correlate with disease progression[16–20]. Even though predominantly expressed on innate immune cells, TLR4 has been found at low levels in activated human and mice CD4 T cells[21,22]. Curiously, TLR4 expression on T cells has been ascribed to both facilitate and inhibit chronic inflammation[23,24], with its role varying according to tissue affected. It remains to be elucidated if TLR4 expression is enriched in CD4$^+$ T cells of RA patients and whether the joint microenvironment engages TLRs directly on CD4$^+$ T cells imprinting dysregulated inflammation and possibly diversifying their pathological function.

The strongest genetic association in RA is with HLA-DR alleles[25]. HLA-DR is constitutively expressed by antigen-presenting cells (APCs), whereupon cognate interaction with the TCR, drives CD4$^+$ T-cell activation[26]. Even though HLA-DR has been used as a marker of activated T cells for more than 40 years[27,28], whether or not HLA-DR expression plays a functional role on activated T cells has remained elusive.

By analyzing freshly obtained synovial fluid and blood from 100 RA patients, we identified a TLR4$^+$ Tfh-like cell population present in blood and expanded in the synovial fluid. Our data unveil that direct TLR4 stimulation functions as a sensor for tissue-damage cues, allowing to spatially tailor the pathological response elicited. Targeting the bidirectional communication between T cells and the synovium microenvironment might be critical to restore joint homeostasis and induce RA remission.

## Results

### A circulating TLR4$^+$CD4$^+$ T-cell population is expanded in the synovial fluid of RA patients.

TLR4 is a robust tissue-damage sensor implicated in RA initiation and progression[18–20,29,30]. Previous studies focused on TLR4 expression by innate immune cells and synoviocytes[19,20,30]. Here, we investigated TLR4 expression by CD4$^+$ T cells in fresh synovial fluid from 12 RA patients

(Supplementary Data 1). Confirming our hypothesis, TLR4 was indeed expressed by ~25% of synovial CD4$^+$ T cells (Fig. 1a). When compared to TLR4$^-$ T cells, TLR4$^+$ T cells, displayed a bigger relative size (FSC-A) and complexity (SSC-A) (Fig. 1b, c). Next, we assessed whether synovial fluid TLR4$^+$ T cells would have a circulating counterpart by examining freshly obtained peripheral blood of 100 RA patients. To ensure that we would be inclusive of CD4$^+$ T cells with higher FSC-A/SSC-A, we gated first on CD3$^{high}$CD4$^{high}$ T cells (Fig. 1d). We could detect two CD4$^+$ T-cell populations with distinct relative sizes and complexities. Through doublet analysis, we observed that these CD4$^+$ T-cell populations distribute along two distinct diagonals, suggesting that they are two distinct populations. As determined for synovial TLR4$^+$ T cells (Fig. 1a), TLR4 expression clustered on FSC-A$^{high}$SSC-A$^{high}$CD4$^+$ T cells (Fig. 1d–f). Circulating TLR4$^+$ T cells frequency (Fig. 1d) ranged between 0.02% and 28.7% (mean 5.89%; mode 1.27%). Donor-matched analysis revealed that TLR4$^+$ T cells were enriched in the synovial fluid (Fig. 1g, h) and correlated with their circulating counterparts (Fig. 1i).

We reasoned that the increase in FSC-A/SSC-A values by synovial fluid and circulating TLR4$^+$ T cells could reverberate their increased activation state. We stained for T-cell activation markers HLA-DR and PD-1. t-SNE analysis showed that PD-1 is expressed by various T-cell populations, including TLR4$^+$ T cells while HLA-DR is selectively expressed by TLR4$^+$ T cells (Fig. 1j). To formally exclude the possibility that bigger size of TLR4$^+$ T cells was owing to cell aggregates, we used HLA-DR as a proxy marker for TLR4$^+$ T cells and sorted HLA-DR$^+$ and HLA-DR$^-$ CD4$^+$ T cells by flow cytometry (Supplementary Fig. 1), labeled sorted cells for CD3 and TLR4 and analyzed them by confocal microscopy (Fig. 1k). Only, HLA-DR$^+$ T cells displayed TLR4 at the cell membrane, where it colocalized with CD3. As FSC-A only provides a relative measure of cell size, we calculated the 3D volume and measured the larger width of both TLR4$^-$ and TLR4$^+$ T cells, and found TLR4$^+$ T cells to be bigger and wider than TLR4$^-$ T cells (Fig. 1k–m). We observed that TLR4$^+$ T cells exhibited membrane projections and alterations in their cell shape. To quantify the latter, we calculated the roundness coefficient, where a roundness index of 1 characterizes perfectly round cells, with values <1 depicting a departure from it[26]. TLR4$^+$ T cells roundness index was ~0.8 (Fig. 1n).

TLR4 expression has been reported on senescent T cells from spondylarthritis patients[31]. To exclude that the cells we identified are non-replicative senescent cells, we labeled them for the proliferation marker Ki-67. We found that ~75% of TLR4$^+$ T cells were undergoing a cell cycle and ~95% upregulated the activation marker CD38 (Fig. 2a–e). Upregulation of HLA-DR, CD38, and Ki-67 by TLR4$^+$ T cells supports their chronic activation, rather than a senescent state.

Collectively, we have identified a previously uncharacterized TLR4$^+$ T-cell population in RA patients. TLR4$^+$ T cells are activated, bigger, highly proliferative, and expanded in the synovial fluid, suggesting a role for these cells as drivers in RA pathology.

### TCR and CD28 stimulation leads to TLR4 upregulation.

To further characterize this TLR4$^+$ T-cell subset, we checked for CD14 co-expression. CD14 was co-expressed by SSC-A$^{high}$ T cells (Fig. 3a). Our microscopy data (Fig. 1k) had shown that TLR4 was indeed expressed by activated HLA-DR$^+$ T cells, raising the possibility that acquisition of innate immune receptors was a consequence of T-cell activation status. To check this hypothesis, we FACS-purified CD4$^+$ T cells resorting to a restrictive lymphocyte gate and by excluding CD14$^+$ cells (Supplementary Fig. 2a). We then stimulated these CD14$^-$CD3$^+$CD4$^+$SSC$^{low}$FSC$^{low}$ T cells through the TCR and

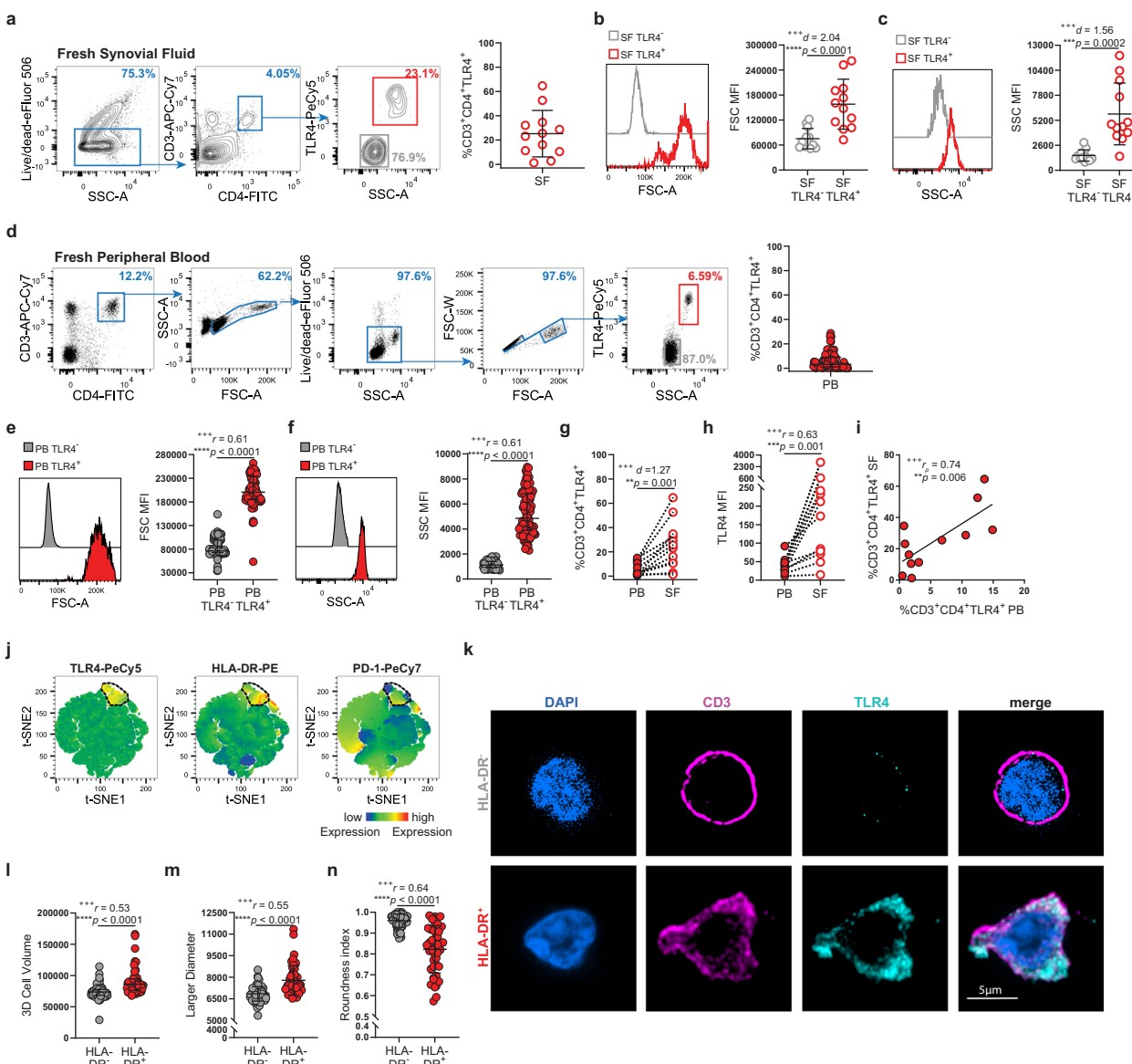

**Fig. 1 RA patients display a circulating TLR4$^+$ T-cell population that is expanded in the synovial fluid. a** Gating strategy and cumulative frequency of CD3$^+$CD4$^+$TLR4$^+$ cells in freshly obtained synovial fluid ($n = 12$ RA patients). **b** Representative histogram and cumulative plot of relative cell size (FSC-A) in TLR4$^-$ (gray) and TLR4$^+$ (red) synovial fluid T cells ($n = 12$ RA patients). **c** Representative histogram and cumulative plot of relative cell complexity (SSC-A) of TLR4$^-$ (gray) and TLR4$^+$ (red) synovial fluid T cells ($n = 12$ RA patients). **d** Gating strategy and cumulative frequency of CD3$^+$CD4$^+$TLR4$^+$ cells in freshly obtained peripheral blood ($n = 100$ RA patients). **e** Representative histogram and cumulative plot of relative cell size (FSC-A) in TLR4$^-$ (gray) and TLR4$^+$ (red) peripheral blood T cells ($n = 100$ RA patients). **f** Representative histogram and cumulative plot of relative cell complexity (SSC-A) of TLR4$^-$ (gray) and TLR4$^+$ (red) peripheral blood T cells ($n = 100$ RA patients). **g** Donor-matched analysis of the frequency of TLR4 expression by CD3$^+$CD4$^+$ T cells in peripheral blood (closed circles; PB) and in synovial fluid (open circles; SF) ($n = 12$ RA patients). **h** Donor-matched analysis of the MFI of TLR4 expression by CD3$^+$CD4$^+$ T cells in peripheral blood (closed circles; PB) and in synovial fluid (open circles; SF) ($n = 11$ RA patients). **i** Correlation between the frequency of CD3$^+$CD4$^+$ TLR4$^+$ T cells in the blood (PB) and in synovial fluid (SF) ($n = 12$ RA patients). **j** t-SNE plots of peripheral blood total CD4$^+$ T cells. The color indicates cell expression levels of labeled markers (TLR4, HLA-DR, and PD-1). Circle demarks TLR4$^+$ cells ($n = 26$ RA patients). **k–n** Confocal microscopy of FACS-purified HLA-DR$^-$ and HLA-DR$^+$ CD4$^+$ T cells. **k** Cells were surface labeled for CD3 and TLR4, stained for DAPI, and analyzed by 3D confocal microscopy. Bar, 5 μm. **l** Cumulative graphs of 3D volume ($n = 76$ HLA-DR$^-$CD4$^+$ T cells; $n = 47$ HLA-DR$^+$CD4$^+$ T cells); **m** larger diameter ($n = 76$ HLA-DR$^-$CD4$^+$ T cells; $n = 47$ HLA-DR$^+$CD4$^+$ T cells), and **n** roundness index ($n = 75$ HLA-DR$^-$CD4$^+$ T cells; $n = 44$ HLA-DR$^+$CD4$^+$ T cells). Data are presented as mean ± SD, for parametric statistical tests, or median ± IQR, for non-parametric statistical tests. Sample normality distribution was tested by using D'Agostino & Pearson normality test. P values ****$p \leq 0.0001$, ***$p \leq 0.001$, **$p \leq 0.01$, *$p \leq 0.05$ were determined by (**b, c, g**) Paired $t$ test; (**e, f, h**) Wilcoxon matched-pairs rank test; (**i**) Pearson Correlation and (**l–n**) Mann–Whitney test. Effect size measures $^{+++}$high, $^{++}$medium, $^+$small were determined by (**b, c, g**) $d$ – Cohen's $d$; (**e, f, h, l–n**) $r$ – correlation coefficient $r$, and (**i**) $r_p$ – Pearson's correlation coefficient.

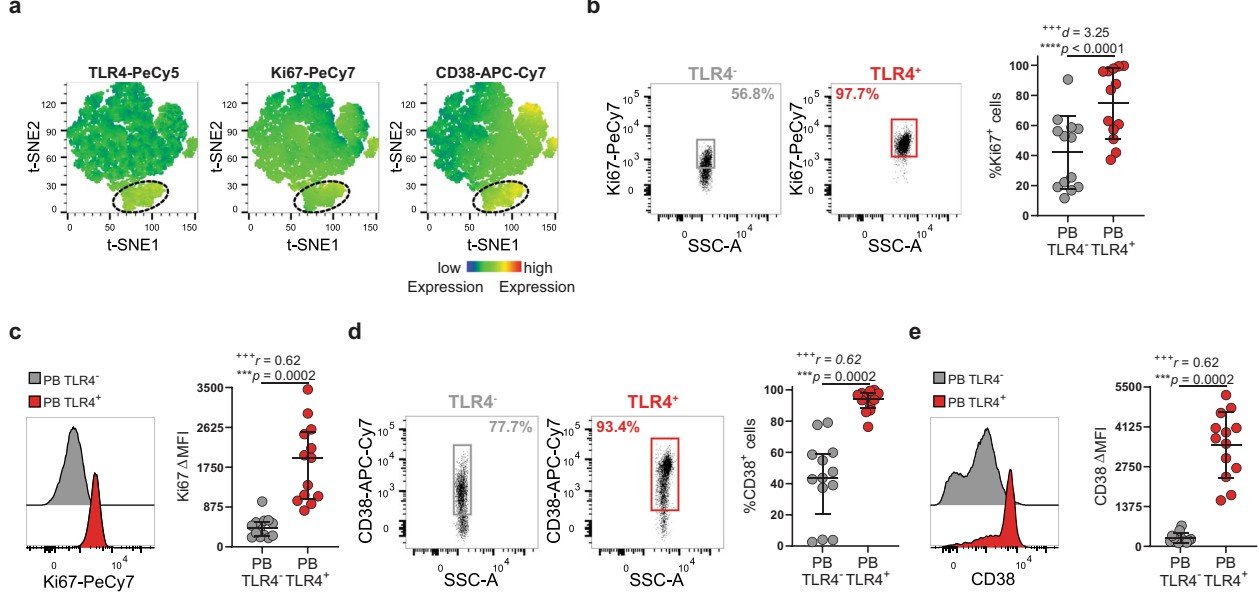

**Fig. 2 TLR4+ T cells display an activated and proliferative phenotype. a** t-SNE plots of peripheral blood total CD3+CD4+ T cells. The color indicates cell expression levels of labeled markers (TLR4, Ki-67, and CD38). Circle demarks TLR4+ cells (*n* = 6 RA patients). **b, c** Representative dot plots and cumulative graphs of the **b** frequency and **c** ΔMFI of Ki-67 expression by TLR4− and TLR4+ peripheral blood T cells (*n* = 13 RA patients). **d, e** Representative dot plots and cumulative graphs of the **d** frequency and **e** ΔMFI of CD38 expression by TLR4− and TLR4+ peripheral blood T cells (*n* = 13 RA patients). ΔMFI was calculated to correct for the distinct autofluorescence of the TLR4− and TLR4+ T-cell populations. ΔMFI was calculated by subtracting the fluorescence intensity minus one (FMO) from median fluorescence intensity (MFI) for each given marker. Data are presented as mean ± SD, for parametric statistical tests, or median ± IQR, for non-parametric statistical tests. Sample normality distribution was tested by using D'Agostino & Pearson normality test. *P* values ****$p \leq 0.0001$, ***$p \leq 0.001$, **$p \leq 0.01$, *$p \leq 0.05$ were determined by (**b**) Paired *t* test and (**c**–**e**) Wilcoxon matched-pairs rank test. Effect size measures +++high, ++medium, +small were determined by (**b**) *d* – Cohen's *d* and (**c**–**e**) *r* – correlation coefficient *r*.

CD28. Sorted CD4+ T cells did not express CD14 (Supplementary Fig. 2b). Nonetheless, as soon as 1 day after TCR and CD28 stimulation CD4+ T cells from RA patients gained CD14 expression, which was further enriched by day 5 (Fig. 3b). To determine whether this TLR4 upregulation was a general feature of activated T cells, we sorted CD3+CD4+ T cells from healthy donors (HD; Supplementary Fig. 3c, d, Supplementary Table 1) and stimulated them through TCR and CD28 for 5, 6 days. Polyclonal T-cell activation led to a homogeneous upregulation of TLR4 in HD. However, its levels were below the detection limits of FACS and could only be observed by microscopy (Fig. 3c, d). TLR4 was functional in T cells from HD, as the addition of LPS to the culture led to an increase in T-cell size (Fig. 3e), survival (Fig. 3f), and proliferation (Fig. 3g, h).

Altogether, our data suggest that TLR4 upregulation is subsequent to sustained TCR and CD28 activation and is not a unique feature of T cells from RA patients. Nonetheless, RA TLR4+ T cells distinguish themselves by their high levels of TLR4 expression.

**TLR4+ T-cell population correlates with anti-CCP antibody titers**. Next, we pursued the relation between TLR4+ T cells and RA demographics, disease presentation, and treatment. TLR4+ T-cell frequency was not affected by age nor sex (Fig. 4a, b). RA has two clinical presentations, seropositive RA in which antibodies to either rheumatoid factor (RF) or to citrullinated (CCP) proteins are present and seronegative RA in which such antibodies are absent. TLR4+ T cells were present in both seropositive and seronegative patients (Fig. 4c–e), with TLR4+ T-cell frequency correlating with anti-CCP antibody titers (Fig. 4f). The majority of patients in our cohort were either in clinical remission (61.7%) or presented low (16.0%) to moderate (21.0%) disease activity. Reflecting the high prevalence of patients with the

controlled disease, we did not detect any correlation between disease activity scores DAS28 ESR (Fig. 4g) and DAS28 CRP (Fig. 4h) and TLR4+ T cells frequency. Likewise, there was no detectable difference in TLR4+ T-cell frequency among treatments (Fig. 4i–k). When analyzed by individual drug use methotrexate (Fig. 4l) and leflunomide (Fig. 4m) exhibited a trend for slightly better and worse outcomes, respectively (Fig. 4l–p). At last, DMARD treatment duration does not impact TLR4+ T-cell frequency (Fig. 4q).

In summary, TLR4+ T cells persist in patients with controlled RA, regardless of the treatment regimen, and correlate with anti-CCP antibody titers.

**HLA-DR drives TLR4 surface expression**. The strongest genetic association for developing RA is carried by HLA-DR alleles[25]. Even though HLA-DR has been used as a marker of T-cell activation for >40 years, its functional role has remained elusive. Intrigued by the strong co-expression between HLA-DR and TLR4 (Figs. 1j, 5a), we analyzed the frequency of TLR4 expression by HLA-DR+CD4+ T cells (Fig. 5b) and reciprocally, the frequency of HLA-DR expression by TLR4+CD4+ T cells (Fig. 5c). Although 87.3% of HLA-DR+CD4+ T cells co-expressed TLR4, 99.4% of TLR4+CD4+ T cells co-expressed HLA-DR. When looking at their cellular abundance, higher expression of HLA-DR was accompanied by greater TLR4 expression (Fig. 5d). Taken together, the above data suggested that there might be a link between HLA-DR and TLR4 expression.

We posited that HLA-DR could control TLR4 expression on T cells. To address this possibility, we purified circulating CD4+ T cells with purity >99% (Supplementary Fig. 3a, b) and incubated them overnight with an anti-HLA-DR-blocking antibody or medium (Fig. 5e). Blocking HLA-DR, led to a stark

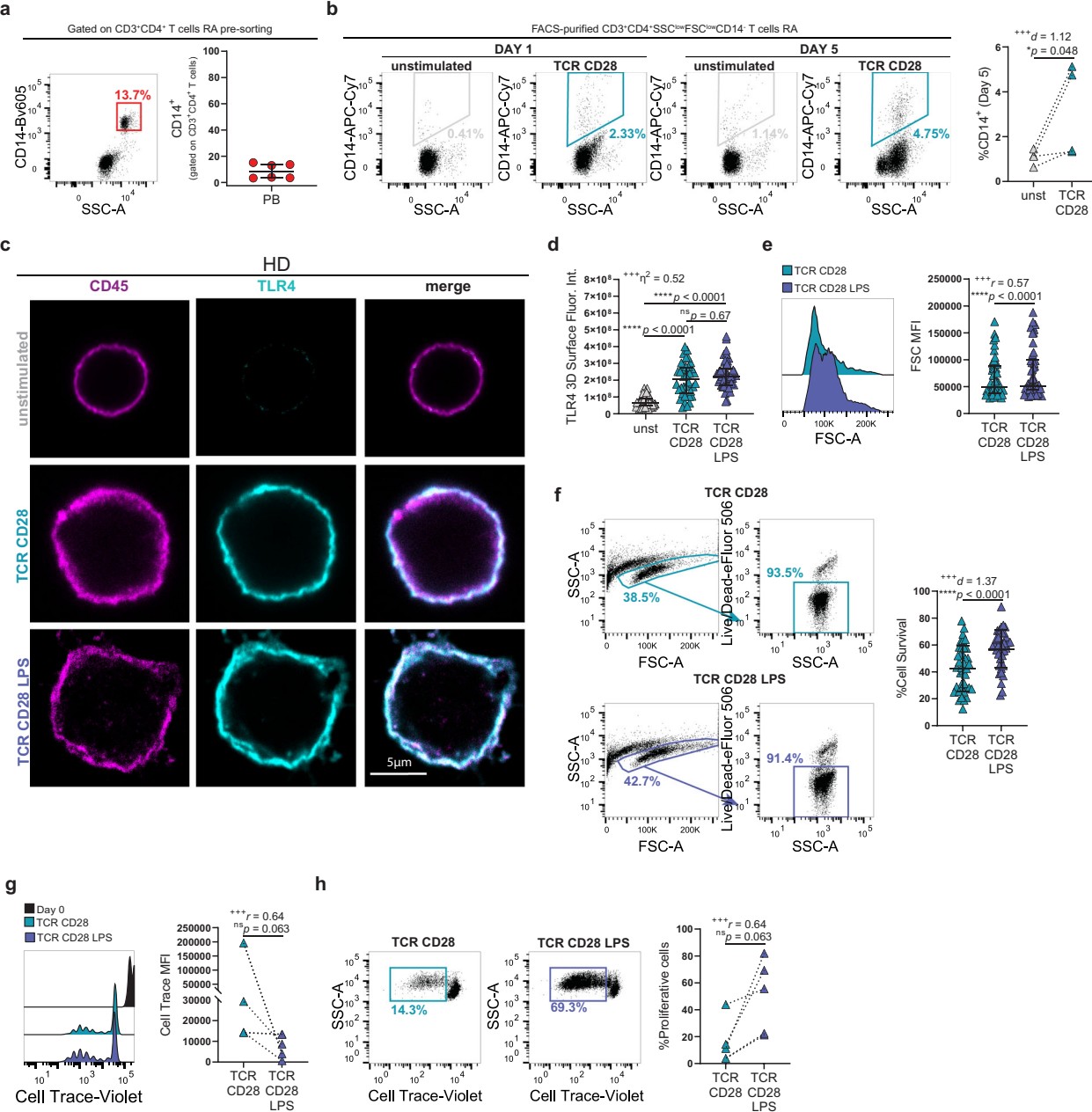

**Fig. 3 TCR and CD28 stimulation lead to CD14 and TLR4 upregulation in RA patients and in healthy donors. a** Representative dot plot and cumulative graph of the frequency of CD14 expression in CD3+CD4+ T cells of peripheral blood of RA patients ($n = 6$ RA patients). **b** Representative dot plot and cumulative graph of CD14 expression by FACS-purified CD14−CD3+CD4+SSClowFSClow T cells from RA patients after 1 and 5 days with and without αCD3 and αCD28 stimulation ($n = 4$ RA patients). **c–d** Confocal microscopy of FACS-purified CD3+CD4+SSClowFSClow T cells from healthy donors (HD) that were either left unstimulated (unst) or were stimulated with αTCR and αCD28 for 5 days in the presence or absence of LPS. **c** Cells were surface labeled for CD45 and TLR4 and analyzed by 3D confocal microscopy. Bar, 5 µm. **d** Cumulative graphs of 3D TLR4 fluorescence ($n = 143$ cells from four different HD, $n = 43$ cells unstimulated, $n = 40$ cells TCR CD28 and $n = 50$ cells TCR CD28 LPS conditions). **e** Representative histogram and cumulative plot of FSC MFI of CD4+ T cells from HD stimulated for 5 days with αTCR and αCD28 in the presence or absence of LPS ($n = 30$ HD, 53 independent experiments). **f** Representative dot plots and cumulative graph of the frequency of viable cells CD4+ T from HD stimulated for 5 days with αTCR and αCD28 in the presence or absence of LPS and labeled with cell viability dye ($n = 30$ HD, 39 independent experiments). **g** Representative histogram and cumulative plot of cell trace MFI of CD4+ T cells from HD stimulated for 5 days with αTCR and αCD28 in the presence or absence of LPS ($n = 5$ HD). **h** Representative dot plots and cumulative graph of proliferative CD4+ T cells from HD stimulated for 5 days with αTCR and αCD28 in the presence or presence or absence of LPS ($n = 5$ HD). Data are presented as mean ± SD, for parametric statistical tests, or median ± IQR, for non-parametric statistical tests. Sample normality distribution was tested by using D'Agostino & Pearson normality test ($n > 6$) or Shapiro–Wilk normality test ($n \leq 6$). $P$ values ****$p \leq 0.0001$, ***$p \leq 0.001$, **$p \leq 0.01$, *$p \leq 0.05$ were determined by (**b**) Ratio-paired $t$ test; (**d**) Kruskal–Wallis test with posttest Dunn's multiple comparisons; the $p$ values are adjusted for multiple comparisons; (**e**, **g**, **h**) Wilcoxon matched-pairs rank test and (**f**) Paired $t$ test. Effect size measures [+++]high, [++]medium, [+]small were determined by (**b**, **f**) $d$ – Cohen's $d$; (**d**) $\eta^2$ – eta-squared and (**e**, **g**, **h**) $r$ – correlation coefficient $r$.

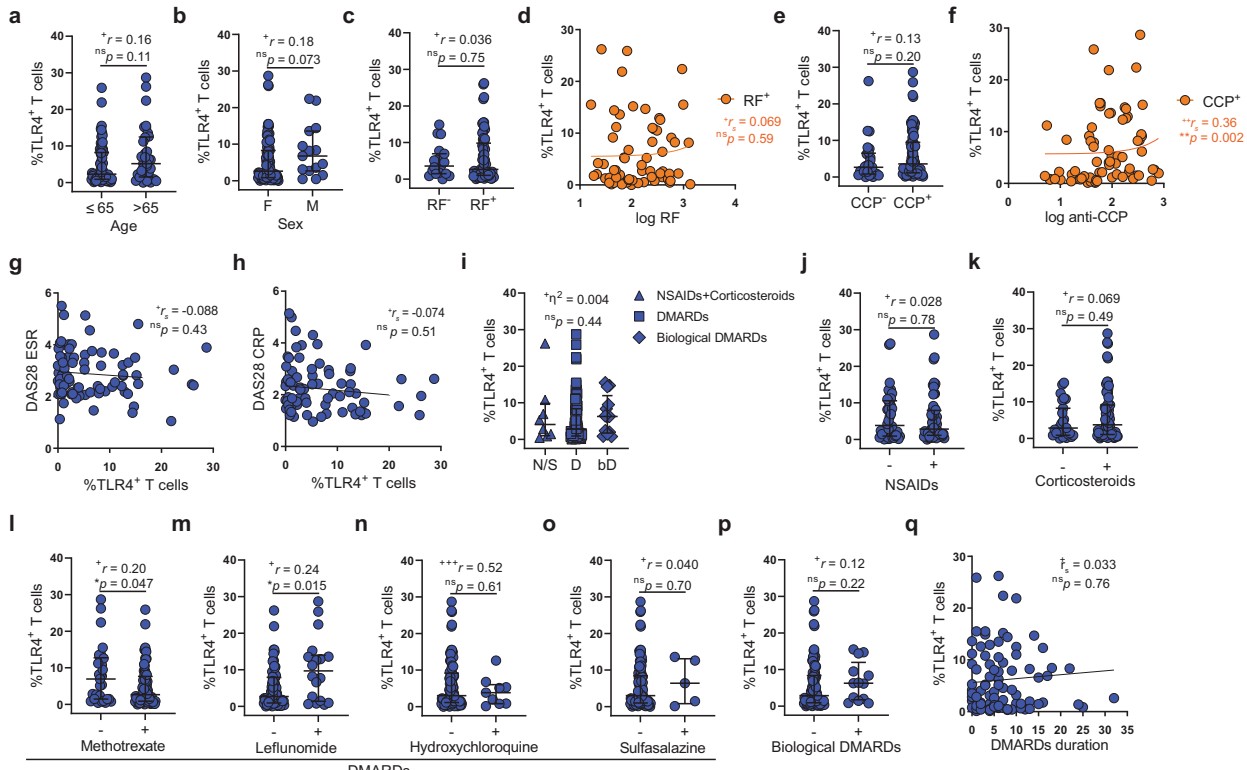

**Fig. 4 The frequency of TLR4+ T correlates with anti-CCP antibody titers and age, independently of treatment. a** Frequency of TLR4+ T cells disaggregated by age ($n = 101$ RA patients; ≤65 years $n = 64$; >65 years $n = 37$). **b** Frequency of TLR4+ T cells disaggregated by sex ($n = 101$ RA patients; female $n = 86$; male $n = 15$). **c** Frequency of TLR4+ T cells disaggregated by factor rheumatoid (RF) status ($n = 84$ RA patients; RF+ $n = 65$; RF− $n = 19$). **d** Correlation between factor rheumatoid titers and frequency of TLR4+ T cells in rheumatoid factor positive patients ($n = 65$ RF+ RA patients). **e** Frequency of TLR4+ T cells disaggregated by factor anti-CCP antibody status ($n = 96$ RA patients; CCP+ $n = 71$; CCP− $n = 25$). **f** Correlation between factor anti-CCP antibody titers and frequency of TLR4+ T cells in CCP-positive patients ($n = 71$ CCP+ RA patients). **g** Correlation between frequency of TLR4+ T cells and DAS28 ESR score ($n = 81$ RA patients). **h** Correlation between frequency of TLR4+ T cells and DAS28 CRP score ($n = 81$ RA patients). **i** Frequency of TLR4+ T cells disaggregated by treatment family (N/S- NSAID and/or corticosteroids $n = 8$; D- DMARDs $n = 80$; bD- biological DMARDs $n = 13$ RA patients). **j–p** Frequency of TLR4+ T cells segregated by medication usage ($n = 101$ RA patients). **j** NSAIDs, **k** corticosteroids, **l** methotrexate, **m** leflunomide, **n** hydroxychloroquine, **o** Sulfasalazine, **p** biological DMARDs. **q** Correlation between DMARD treatment duration and frequency of TLR4+ T cells ($n = 89$ RA patients). Data are presented as mean ± SD, for parametric statistical tests, or median ± IQR, for non-parametric statistical tests. Sample normality distribution was tested by using D'Agostino & Pearson normality test ($n > 6$) or Shapiro–Wilk normality test ($n ≤ 6$). P values ****$p ≤ 0.0001$, ***$p ≤ 0.001$, **$p ≤ 0.01$, *$p ≤ 0.05$ were determined by (**a–c**, **e**, **j–p**) Mann–Whitney test; (**d**, **f–h**, **q**) Spearman Correlation and (**i**) Kruskal–Wallis test. Effect size measures +++high, ++medium, +small were determined by (**a–c**, **e**, **j–p**) $r$ – correlation coefficient $r$; (**d**, **f–h**, **q**) $r_s$ – Spearman's correlation coefficient and (**i**) $\eta^2$ – eta-squared.

decrease in TLR4 surface expression (Fig. 5e), indicating that HLA-DR regulates TLR4 expression.

Our data identify, for the first time, a functional role for HLA-DR on CD4+ T cells through the regulation of TLR4 expression and suggests a novel mechanism by which HLA-DR might drive RA disease susceptibility.

**TLR4+ T cells share features of Tfh cells**. Tfh-like T cells have been implicated in RA and other chronic inflammatory diseases owing to their capability to induce antibody production[5,32,33]. We checked whether TLR4+ T cells would share Tfh features, namely high expression of chemokine receptor CXCR5 and of the co-receptors PD-1 and ICOS. The receptor gating for each T-cell population was established by their individual fluorescence minus one (FMO) value (Supplementary Fig. 4). Even though CXCR5 (Fig. 6a–c) and PD-1 (Fig. 6a, d, e) could be detected in both TLR4− and TLR4+ T-cell populations, they were enriched in TLR4+ T cells with a co-expression of ~60% (Fig. 6f). Curiously, ICOS was more expressed in TLR4− than in TLR4+ T cells (Fig. 6a, g, h). Nonetheless, in TLR4+ T cells co-expression of ICOS and CXCR5 (Fig. 6i) and ICOS and PD-1 (Fig. 6j) was

enriched. The fact that TLR4+ T cells are enriched in CXCR5 and PD-1 suggests that they might consist of a circulating Tfh-like population[34,35]. To characterize this further, we explored whether the enrichment in TLR4+ T cells could reflect the frequency of circulating Tfh cells. TLR4+ T-cell frequency positively correlated with the frequency of CXCR5+ (Fig. 6k) and PD-1+ (Fig. 6l) circulating CD4+ T cells.

Altogether, these data indicate that TLR4+ T cells display Tfh-like features.

**TLR4+ T cells display migratory phenotype to inflamed tissues**. TLR4+ T-cell enrichment in synovial fluid (Fig. 1g, h) cannot be fully explained by their CXCR5 expression. Therefore, we checked for the expression of chemokine receptors CCR2 and CCR6 that regulate T-cell migration to inflamed tissues and whose ligands are abundantly present in arthritic synovium and have been implicated in the disease[36,37]. Both CCR2 and CCR6 were upregulated by TLR4+ T cells (Fig. 7a–e). CCR2 and CCR6 are expressed by ~100% and ~30% of TLR4+ T cells, respectively (Fig. 7b–e). Although CCR2 guides a broad range of immune cells into sites of inflammation, CCR6 is associated with the recruitment of IL-17-producing

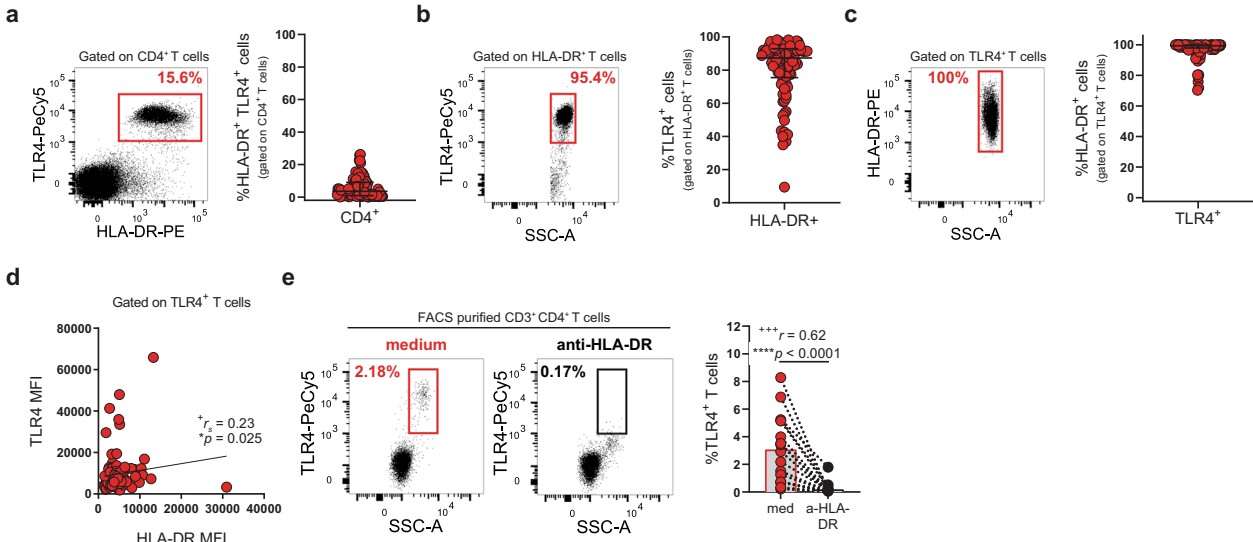

**Fig. 5 Blocking HLA-DR abrogates TLR4 surface expression in T cells. a** Representative plot and cumulative graph ($n = 99$ RA patients) of the frequency of HLA-DR⁺TLR4⁺ T cells. **b** Representative plots and cumulative graph ($n = 99$ RA patients) of the frequency of TLR4 expression by HLA-DR⁺ T cells. **c** Representative plots and cumulative graph ($n = 99$ RA patients) of the frequency of HLA-DR expression by TLR4⁺ T cells. **d** Correlation between HLA-DR and TLR4 MFIs in TLR4⁺ T cells ($n = 99$ RA patients). **e** Representative plots and cumulative graph ($n = 17$ RA patients) of the frequency of TLR4⁺ T cells after incubating FACS-purified CD4⁺ T cells with a blocking antibody to HLA-DR for 18 hours. Data are presented as mean ± SD, for parametric statistical tests, or median ± IQR, for non-parametric statistical tests. Sample normality distribution was tested by using D'Agostino & Pearson normality test. *P* values ****$p \leq 0.0001$, ***$p \leq 0.001$, **$p \leq 0.01$, *$p \leq 0.05$ were determined by (**d**) Spearman correlation and (**e**) Wilcoxon matched-pairs rank test. Effect size measures ⁺⁺⁺high, ⁺⁺medium, ⁺small were determined by (**d**) $r_s$ – Spearman's correlation coefficient and (**e**) $r$ – correlation coefficient $r$.

T cells to inflamed joints[38], suggesting an IL-17 inflammatory component to TLR4⁺ T-cell synovial recruitment. To address this possibility, we checked whether TLR4⁺ T cells upregulate receptors for pro-inflammatory cytokines that are overexpressed in inflamed synovium (IL-1, IL-6, and IL-17) and which have been implicated in IL-17 production[38–40] (Figs. 7a, 8a, b). IL-1R was selectively upregulated by TLR4⁺ T cells (Figs. 7a, 8c, d). As expected from IL-6 pleiotropic role, IL-6R was similarly expressed by both TLR4⁺ and TLR4⁻ T-cell populations (Fig. 8a, e, f). Finally, IL-17R was greatly enriched in TLR4⁺ T cells (Fig. 8b, g, h). In addition, IL-2 alpha (IL-2Rα), which regulates cell size was upregulated by TLR4⁺ T cells (Fig. 6b, i, j).

Taken together, TLR4⁺ T cells emerge as a Tfh-like cell population with a preferential tropism for inflamed tissues and increased capability to respond to IL-17-promoting stimuli IL-1 and IL-17.

**TLR4 engagement reprograms TLR4⁺ T-cell inflammatory profile**. In experimental autoimmune encephalitis, TLR4 engagement on CD4⁺ T cells has been reported to function as a co-receptor boosting T-cell survival and proliferation without affecting the amount of the cytokines produced[24]. Whether or not direct TLR4 engagement on human CD4⁺ T cells modulates or alters CD4⁺ T-cell inflammatory profile has remained unanswered.

To unveil the contribution of direct TLR4 engagement on T-cell inflammatory profile, we purified circulating CD4⁺ T cells (Supplementary Fig. 3b; purity >99%) and stimulated them with TLR4 ligand LPS in the presence or absence of TCR and ICOS engagement. We looked at antibodies (IL-21[32] and IL-10[41]), or and joint damage (IL-10[42], IL-17[8,12,43,44], and TNF-α[45]) inducing cytokines. Circulating TLR4⁺ T cells produced IL-10, IL-21, and IL-17 in unstimulated conditions, supporting their ongoing activation state. In vitro, IL-21 production required TCR and ICOS stimulation and was completely non-responsive to LPS (Fig. 9a, b). In contrast, LPS, in combination with TCR and ICOS

stimulation, boosted IL-10, IL-17, and TNF-α production (Fig. 9c–h). Moreover, LPS alone was sufficient to drive the production of IL-10 and trended to increase IL-17 and TNF-α production (Fig. 9c, g).

Altogether these data indicate that direct TLR4 stimulation goes beyond functioning as a co-receptor boosting TCR-driven response and suggest that TLR4 engagement by LPS might reprogram TLR4⁺ T cells from an IL-21 driven pro-antibody to an inflammatory program fueling joint damage

**Direct recognition of TLR4 ligands present in synovial fluid drives IL-17 production, independently of antigen recognition**. Increased expression of endogenous TLR4 ligands has been observed in the blood and synovial fluid of RA patients, with a role in arthritis being suggested in mice models[46–49]. Of all the proposed endogenous TLR4 ligands, tenascin-C (TNC) is the one more thoroughly analyzed, including the molecular identification of its binding sites on TLR4[50]. We quantified TNC in synovial fluid of RA patients (Fig. 10a–c). Synovial TNC levels are independent of the duration of DMARD treatment (Fig. 10a). Moreover, TLR4⁺ T cells are enriched in synovial fluids with higher TNC levels (Fig. 10b, c), opening the possibility that TNC might play a role in the enrichment of TLR4⁺ T cells in the synovial fluid. As circulating TLR4⁺ T cells produced IL-17 and IL-10 prior to in vitro restimulation (Fig. 9), we wondered whether this basal cytokine production was due to the ongoing engagement of TLR4. To address this possibility, we treated circulating TLR4⁺ T cells with either medium or with the TLR4 signaling inhibitor CLI-095. Blocking TLR4 signaling hampered both IL-17 and IL-10 production (Supplementary Fig. 5a–d). To further explore the role of direct TLR4 engagement by synovial components, we stimulated sorted CD3^high^CD4^high^ T cells with either TNC (Supplementary Fig. 5h, i) or with cell-depleted synovial fluid in the presence or absence of TLR4 signaling inhibitor (Fig. 10d–g). Whereas stimulation with TNC led to the production of IL-10 and TNF-α, but not of IL-17,

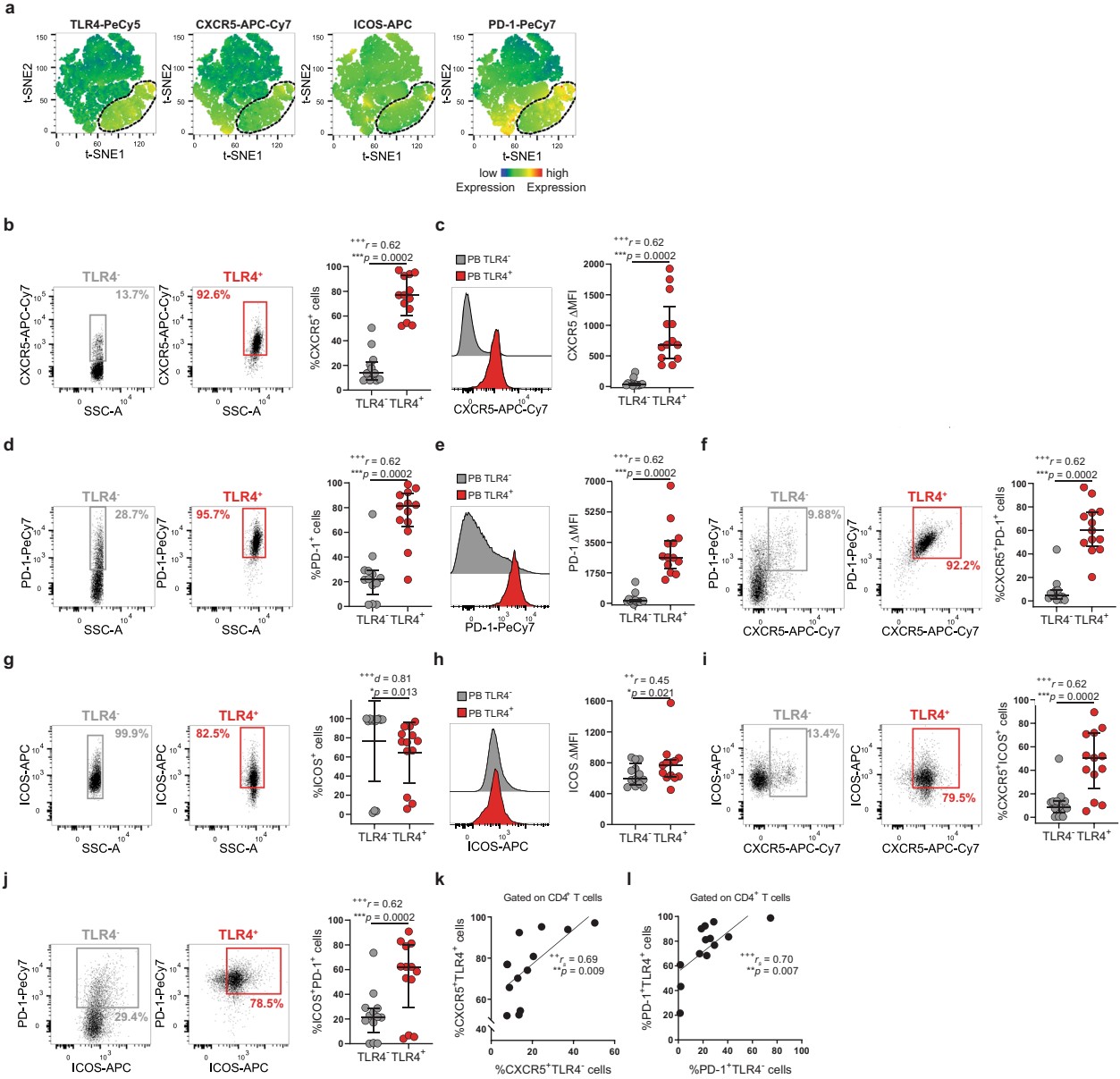

**Fig. 6 TLR4+ T cells have features of Tfh-like cells. a** t-SNE plots of peripheral blood total CD4+ T cells. The color indicates cell expression levels of labeled markers (TLR4, CXCR5, ICOS, and PD-1). Circle demarks TLR4+ cells (*n* = 6 RA patients). **b, c** Representative plots and cumulative analysis (*n* = 13 RA patients) of CXCR5 **b** frequency and **c** ΔMFI in TLR4+ (red) versus TLR4− (gray) T cells. **d, e** Representative plots and cumulative analysis (*n* = 13 RA patients) of PD-1. **d** frequency and **e** ΔMFI in TLR4+ (red) versus TLR4− T cells (gray). **f** Representative plots and cumulative analysis (*n* = 13 RA patients) of the frequency of CXCR5 and PD-1 co-expression TLR4+ (red) versus TLR4− (gray) T cells. **g, h** Representative plots and cumulative analysis (*n* = 13 RA patients) of ICOS **g** frequency and **h** ΔMFI in TLR4+ (red) versus TLR4− (gray) T cells. **i** Representative plots and cumulative analysis (*n* = 13 RA patients) of the frequency of CXCR5 and ICOS co-expression in TLR4+ (red) versus TLR4− (gray) T cells. **j** Representative plots and cumulative analysis (*n* = 13 RA patients) of the frequency of ICOS and PD-1 co-expression TLR4+ (red) versus TLR4− (gray) T cells. **k** Correlation between the frequency of TLR4+CXCR5+ T cells and TLR4−CXCR5+ cells (*n* = 13 RA patients). **l** Correlation between the frequency of TLR4+PD1+ T cells and TLR4−PD1+ cells (*n* = 13 RA patients). ΔMFI was calculated to correct for the distinct autofluorescence of the TLR4− and TLR4+ T-cell populations. ΔMFI was calculated by subtracting the fluorescence intensity minus one (FMO) from median fluorescence intensity (MFI) for each given marker. Data are presented as mean ± SD, for parametric statistical tests, or median ± IQR, for non-parametric statistical tests. Sample normality distribution was tested by using D'Agostino & Pearson normality test. *P* values ****p ≤ 0.0001, ***p ≤ 0.001, **p ≤ 0.01, *p ≤ 0.05 were determined by (**b–f**, **h–j**) Wilcoxon matched-pairs rank test; (**g**) Paired *t* test and (**k, l**) Spearman Correlation. Effect size measures +++high, ++medium, +small were determined by (**b–f**, **h–j**) *r* – correlation coefficient *r*; (**g**) *d* − Cohen's *d*; and (**k, l**) *r*s – Spearman's correlation coefficient.

IL-21, or IFN-γ (Supplementary Fig 5h–I), stimulation with-synovial fluid-induced IL-17 in all the 5 donors analyzed with increases ranging between 1.5- and 4.7-fold (Fig. 10d). In comparison with LPS stimulation (Fig. 9), cell-depleted synovial fluid was less prone to induce IL-10 and TNF-α production with

only three out of five donors responding to stimulation (Fig. 10e, f) and uncapable of inducing IL-21 production (Fig. 10g). Notably, increased IL-17 production appears to be mediated by direct TLR4 engagement, as the addition of TLR4 specific signaling inhibitor CLI-095 at least impairs (two

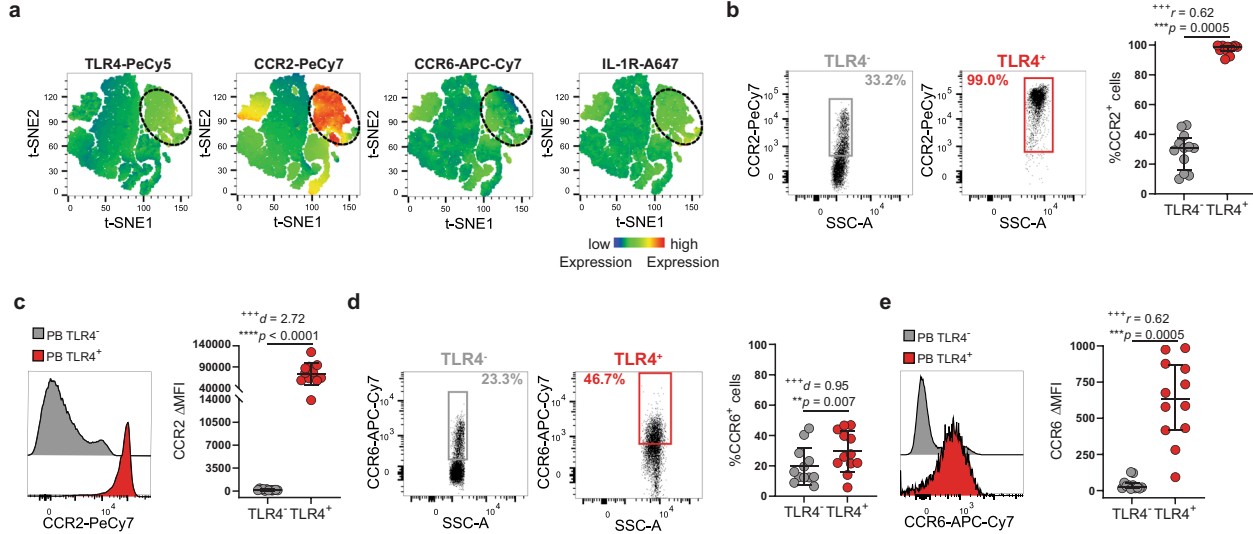

**Fig. 7 TLR4+ T cells express inflammatory chemokine receptors CCR2 and CXCR6. a** t-SNE plots of peripheral blood total CD4+ T cells. The color indicates cell expression levels of labeled markers (TLR4, CCR2, CCR6, IL-1R). Circle demarks TLR4+ cells (n = 6 RA patients). **b, c** Representative plots and cumulative graph (n = 12 RA patients) of CCR2 **b** frequency and **c** ΔMFI in TLR4+ (red) and TLR4− (gray) T cells. **d, e** Representative plots and cumulative graph (n = 12 RA patients) of CCR6 **d** frequency and **e** ΔMFI in TLR4+ (red) and TLR4− (gray) T cells. ΔMFI was calculated to correct for the distinct autofluorescence of the TLR4− and TLR4+ T-cell populations. ΔMFI was calculated by subtracting the fluorescence intensity minus one (FMO) from median fluorescence intensity (MFI) for each given marker. Data are presented as mean ± SD, for parametric statistical tests, or median ± IQR, for non-parametric statistical tests. Sample normality distribution was tested by using D'Agostino & Pearson normality test. P values ****p ≤ 0.0001, ***p ≤ 0.001, **p ≤ 0.01, *p ≤ 0.05 were determined by (**b, e**) Wilcoxon matched-pairs rank test and (**c, d**) Paired t test. Effect size measures +++high, ++medium, +small were determined by (**b, e**) r – correlation coefficient r and (**c, d**) d – Cohen's d.

out of five donors), and in most cases (three out of five donors) completely abrogates, IL-17 production (Fig. 10d). In contrast to LPS (Fig. 9), direct TLR4 engagement by endogenous synovial ligands boosted IL-17 production independently of TCR cross-linking (Fig. 10d).

To scope the pathophysiological role that endogenous TLR4 ligands might exert on the inflammatory program of synovial TLR4+ T cells, we compared the cytokine profile of circulating and synovial TLR4+ T cells ex vivo. In this approach, freshly obtained and paired blood and synovial fluid mononuclear cells were immediately labeled for IL-17, IL-10, TNF-α, and IL-21 (Fig. 10h–k). In four out of five donors, ex vivo IL-17 production by TLR4+ T cells tended to be higher in the synovial fluid than in the blood (Fig. 10h). Curiously, IL-10 production appears to be less compartmentalized with only one donor (out of five) displaying higher IL-10 production by synovial TLR4+ T cells (Fig. 10i). At last, in our sampling TNF-α and IL-21 production was hard to detect in either blood or synovial TLR4+ T cells ex vivo (Fig. 10j, k).

Altogether, our results indicate that direct TLR4 engagement by endogenous ligands in synovial fluid favors the production of IL-17. In contrast with LPS, endogenous synovial TLR4 ligands reprogram TLR4+ T cells inflammatory profile independently of TCR engagement. Lastly, cytokine production by synovial TLR4+ T cells suggests a major role for IL-17 in their pathogenic function.

## Discussion

RA is a chronic inflammatory disease where CD4+ T cells and joint tissue dysregulation synergize in propagating chronic inflammation and articular destruction. Treatment of RA remains challenging as the identity of CD4+ T-cell population driving RA and the mechanism by which joint microenvironment impinges dysregulated T-cell activation remain elusive. Here, we identified a circulating TLR4+ T-cell population that is enriched in synovial

fluid of RA patients. TLR4+ T cells are uniquely attuned to respond distinctively to different contextual clues by reconciling an ability to potentially promote systemic antibody production with an in situ synovial driven tissue-damage program. Our results highlight the contribution of spatial compartmentalization to T-cell-driven pathogenicity and the role of tissue environment in tailoring site-specific T-cell responses.

Tfh-like cell populations have been described in several chronic inflammatory diseases including rheumatoid arthritis[2], lupus nephritis[51], and systemic sclerosis[45]. In addition, in RA, a population of IL-21-producing peripheral helper T (Tph) cells has been identified[32]. Here, we have identified a previously unknown Tfh-like population. TLR4+ T cells were enriched in Tfh cell markers, CXCR5, and PD-1[34,35,52], and their frequency in circulation correlated with anti-CCP antibody levels. These sets of Tfh/Tph cells might indeed account for distinct T-cell populations or might represent the same cell population in different disease stages and/or responses to treatment. Distinctly from the previous reports[2,32,45,51], we analyzed freshly obtained blood and synovial fluid samples, rather than frozen ones. Fresh samples facilitate the identification of infrequent T-cell populations and the detection of certain markers and allow for better detection of changes in cell size and shape.

Early descriptions of TLR4+ T had similarly reported an increase in cell size[41]. Likewise, in vitro and in vivo experiments show that IL-17-producing cells have a bigger size which has been associated with increased cytokine secretion in vitro[42]. As TLR4+ T cells FSC-A values were outside the conventional lymphocyte gate, we took care to exclude the occurrence of cell aggregates[53]. First, our doublet analysis (FSC-W vs FCS-A) into two distinct diagonals is suggestive of two cell populations rather than doublets. Second, confocal microscopy of purified CD4+ T cells (~99% purity) confirmed co-expression of TLR4 and CD3 exclusively by HLA-DR+FSC-Ahigh cells. TLR4 was expressed uniformly along the cell membrane, excluding the possibility of TLR4 acquisition through trogocytosis subsequent to prior

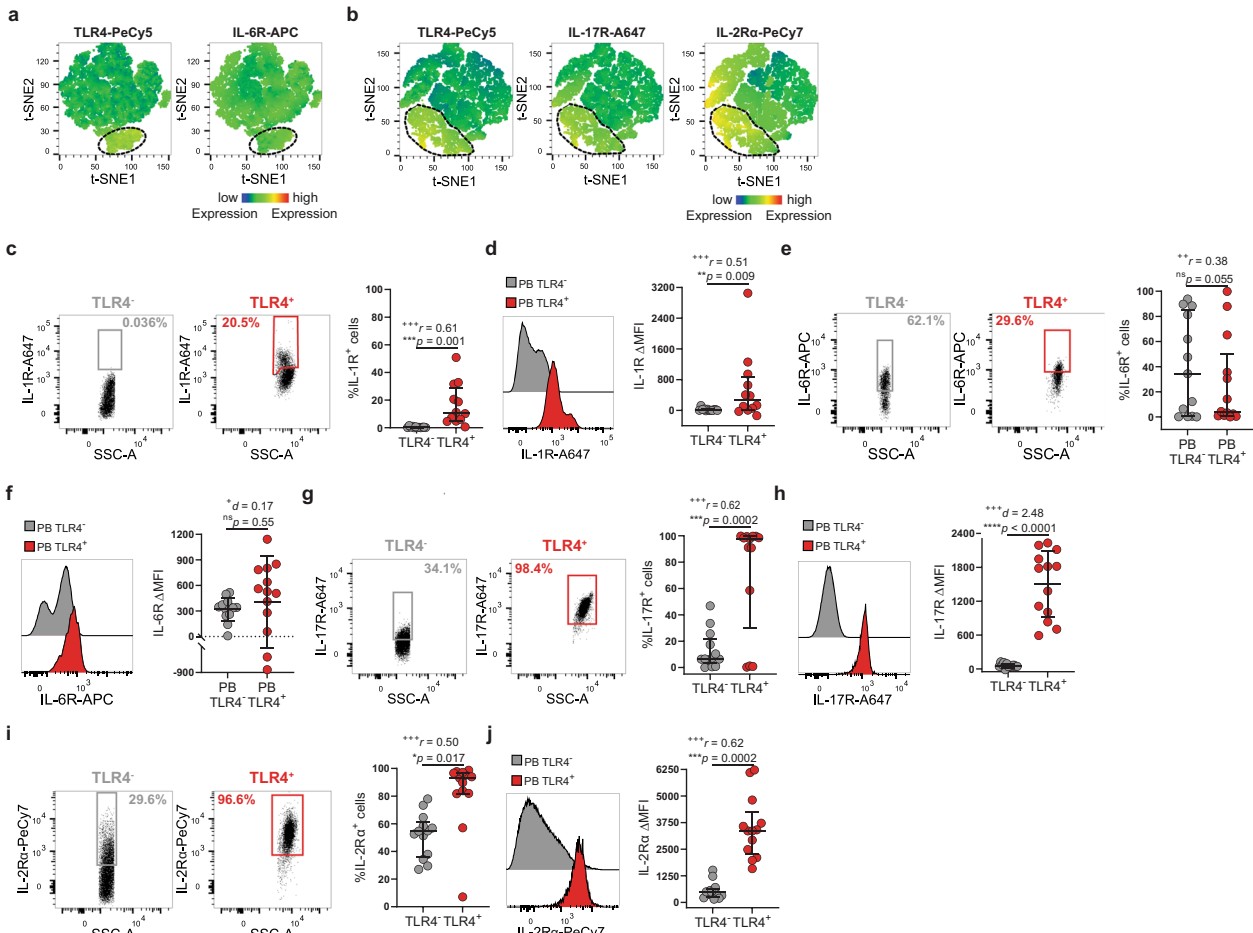

**Fig. 8 TLR4+ T cells upregulate receptors for inflammatory chemokine cytokines. a, b** t-SNE plots of peripheral blood total CD4+ T cells. The color indicates cell expression levels of the labeled marker. **a** TLR4, IL-6R; TLR4, IL-17R, and IL-2Rα. Circle demarks TLR4+ cells (n = 6 RA patients). **c, d** Representative plots and cumulative graph (n = 12 RA patients) of IL-1R **c** frequency and **d** ΔMFI in TLR4+ (red) and TLR4− (gray) T cells. **e, f** Representative plots and cumulative graph (n = 13 RA patients) of IL-6R **e** frequency and **f** ΔMFI in TLR4+ (red) and TLR4− (gray) T cells. **g, h** Representative plots and cumulative graph (n = 13 RA patients) of IL-17R **g** frequency and **h** ΔMFI in TLR4+ (red) and TLR4− (gray) T cells. **i–j** Representative plots and cumulative graph (n = 13 RA patients) of IL-2Rα **i** frequency and **j** ΔMFI in TLR4+ (red) and TLR4− (gray) T cells. ΔMFI was calculated to correct for the distinct autofluorescence of the TLR4− and TLR4+ T-cell populations. ΔMFI was calculated by subtracting the fluorescence intensity minus one (FMO) from median fluorescence intensity (MFI) for each given marker. Data are presented as mean ± SD, for parametric statistical tests, or median ± IQR, for non-parametric statistical tests. Sample normality distribution was tested by using D'Agostino & Pearson normality test. P values ****p ≤ 0.0001, ***p ≤ 0.001, **p ≤ 0.01, *p ≤ 0.05 were determined by (**c–e, g, i, j**) Wilcoxon matched-pairs rank test and (**f, h**) Paired t test. Effect size measures +++high, ++medium, +small were determined by (**c–e, g, i, j**) r – correlation coefficient r and (**f, h**) d – Cohen's d.

interactions with APCs[54]. The increase in cell size combined with membrane projections likely underpins the twofold increase in FSC-A value detected by flow cytometry. An increase in cell size accompanied by the expression of activation markers CD38 and HLA-DR further argue that TLR4+ T cells are indeed blasts. To further characterize whether the acquisition of TLR4 was related to T-cell activation, we checked if T cells from RA patients could acquire the TLR4 co-receptor, and often used monocyte marker, CD14. To this end, we sorted CD14−CD3+CD4+SSC^low^FSC^low^ cells and stimulated them for 5 days through TCR and CD28. T cells from RA patients efficiently acquired CD14 in a TCR stimulation-dependent manner. Similarly, we sorted CD4+ T cells from HDs and observed that TCR and CD28 stimulation led to TLR4 expression, albeit at lower levels than the one observed in RA patients. Nonetheless, TLR4 in HD T cells was functional and mediated increased cell size, survival, and proliferative capacity. Altogether these data support the view that TCR stimulation drives TLR4 expression in T cells and that in HDs TLR4 levels are contained, whereas in RA patients this regulatory threshold seems

to be breached. It is tempting to speculate that higher TLR4 levels might be capable of driving T-cell reprogramming in an antigen recognition-independent manner, while lower TLR4 levels are restricted to driving T-cell survival and proliferation.

HLA-DR is class II major histocompatibility molecules (MHC II) commonly present in APCs, where recognition of foreign-antigen bearing MHC by their cognate TCR on T cells drives antigen-specific T-cell activation[26,55]. HLA-DR haplotypes constitute the strongest genetic association with RA[56]. So far, research addressing this genetic association has focused on identifying the immunodominant peptide presented by HLA-DR on APCs driving dysregulated T-cell activation in RA patients. Even though, several CCP candidate peptides can be presented by HLA-DRB1[57], the search for immunodominant T-cell epitopes has so far revealed unfruitful. The observation that HLA-DR is expressed by activated T cells is longstanding[27,28], including a recent identification of an HLA-DR+ T-cell subset in RA patients[58]. Nonetheless, the function of HLA-DR in T cells has remained enigmatic.

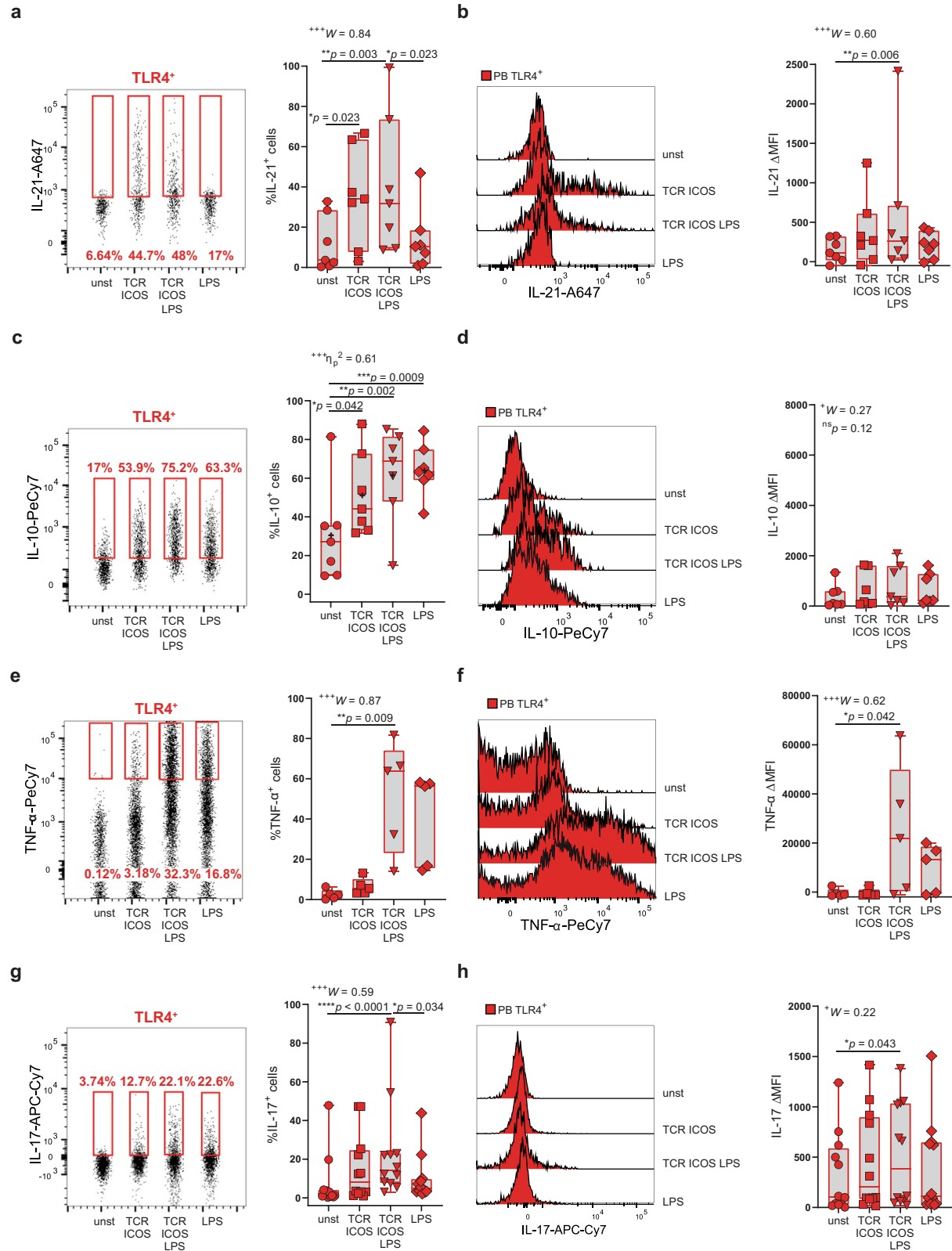

We unveiled for the first time a function for HLA-DR on T cells. By using an anti-HLA-DR-blocking antibody on FACS-purified CD4[+] T cells, we uncovered that HLA-DR regulates TLR4 surface expression on T cells. Previous works have reported that HLA-DRB1 might interact with citrullinated calreticulin at the surface of cells[59]. It is possible that this is the mechanism underpinning TLR4 regulation by HLA-DR. However, since we

show that TCR stimulation drives TLR4 and CD14 expression in CD4[+] T cells from HDs and RA patients, we favor the view that HLA-DR:TCR interactions between neighboring T cells underpins TLR4 upregulation. Non-cognate HLA-DR:TCR interactions between APCs and T cells are known to alter T-cell genetic profile[60,61]. Thus, it is possible that T-T-cell interactions through non-cognate HLA-DR:TCR contacts might drive TLR4 gene

**Fig. 9 Direct recognition of LPS by TLR4⁺ T cells reprograms their cytokine program.** FACS-purified CD3^high^CD4^high^ T cells from freshly obtained peripheral blood were cultured for 18 hours and stimulated with either α-CD3 and α-ICOS (TCR ICOS); α-CD3, α-ICOS, and LPS (TCR ICOS LPS); LPS alone; or left unstimulated (unst). **a** Frequency and **b** ΔMFI of IL-21 production by TLR4⁺ T cells ($n = 7$ RA patients). **c** Frequency and **d** ΔMFI of IL-10 production by TLR4⁺ T cells ($n = 7$ RA patients). **e** Frequency and **f** ΔMFI of TNF-α production by TLR4⁺ T cells ($n = 5$ RA patients). **g** Frequency and **h** ΔMFI of IL-17 production by TLR4⁺ T cells ($n = 12$ RA patients). ΔMFI was calculated by subtracting the fluorescence intensity minus one (FMO) from median fluorescence intensity (MFI) for each given marker. Data are presented as median ± IQR and mean (+), for parametric statistical tests, median ± IQR, for non-parametric statistical tests. Sample normality distribution was tested by using D'Agostino & Pearson normality test ($n > 6$) or Shapiro–Wilk normality test ($n ≤ 6$). P values ****$p ≤ 0.0001$, ***$p ≤ 0.001$, **$p ≤ 0.01$, *$p ≤ 0.05$ were determined by (**a**, **b**, **d**, **e–h**) Friedman test with posttest Dunn's multiple comparisons and (**c**) Repeated measures ANOVA with posttest Tukey's multiple comparisons; the p values are adjusted for multiple comparisons. Effect size measures ⁺⁺⁺high, ⁺⁺medium, ⁺small were determined by (**a**, **b**, **d–h**) W – Kendall's W and (**c**) $\eta_p^2$ – partial eta-squared.

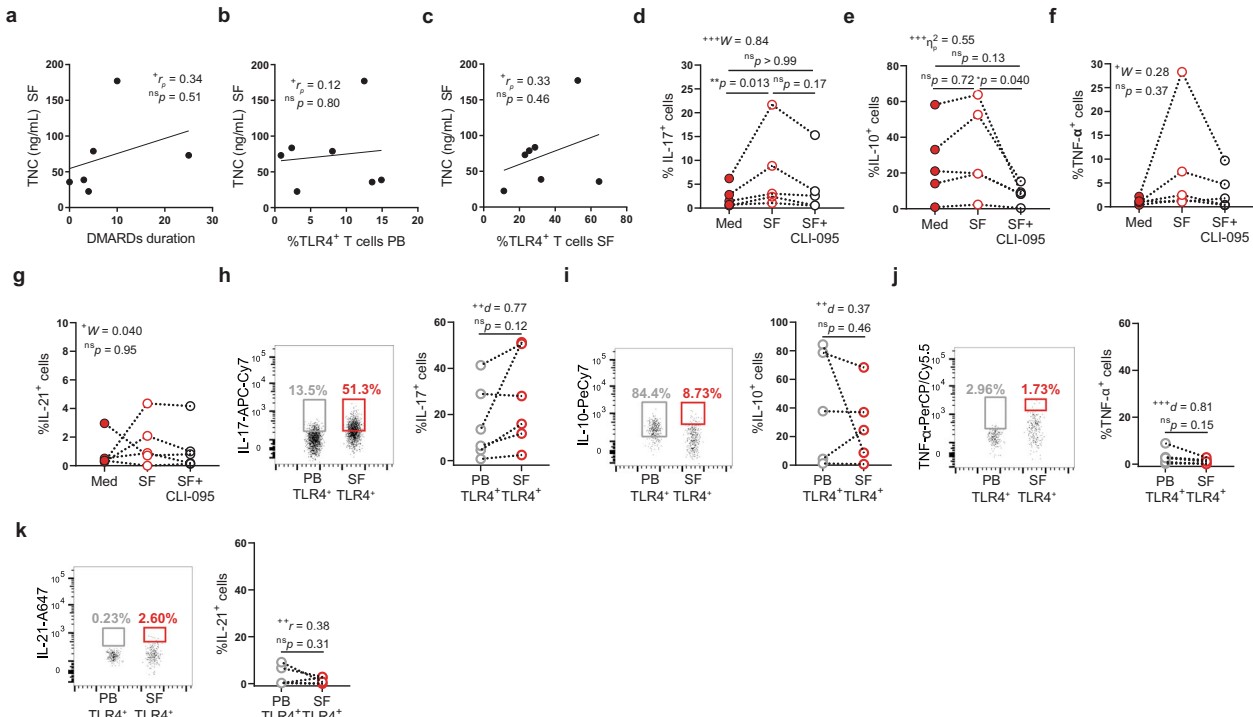

**Fig. 10 Direct recognition of TLR4 ligands present in synovial fluid drives IL-17 production, independently of antigen recognition. a–c** Correlation between synovial fluid tenascin-C levels and **a** DMARD duration ($n = 6$ RA patients), **b** frequency of circulating (PB) TLR4⁺ T cells ($n = 7$ RA patients), and **c** frequency of synovial fluid (SF) TLR4⁺ T cells ($n = 7$ RA patients). **d–g** FACS-purified CD3^high^CD4^high^ T cells from peripheral blood were cultured for 18 hours in the presence of medium (Med), synovial fluid (SF), or TLR4 signaling inhibitor (CLI-095). Frequency of **d** IL-17, **e** IL-10, **f** TNF-α, and **g** IL-21 production by TLR4⁺ T cells ($n = 5$ RA patients). **h–k** Ex vivo production of **h** IL-17 ($n = 6$ RA patients), **i** IL-10 ($n = 5$ RA patients), **j** TNF-α ($n = 5$ RA patients) and **k** IL-21 ($n = 5$ RA patients) by TLR4⁺ T cells in freshly obtained peripheral blood (PB) and synovial fluid (SF) donor paired samples. ΔMFI was calculated by subtracting the fluorescence intensity minus one (FMO) from median fluorescence intensity (MFI) for each given marker. FMOs were calculated independently for blood and synovial fluid FACS analysis. Data are presented as mean ± SD, for parametric statistical tests, or median ± IQR, for non-parametric statistical tests. Sample normality distribution was tested by using D'Agostino & Pearson normality test ($n > 6$) or Shapiro–Wilk normality test ($n ≤ 6$). P values ****$p ≤ 0.0001$, ***$p ≤ 0.001$, **$p ≤ 0.01$, *$p ≤ 0.05$ were determined by (**a–c**) Pearson correlation; (**d**, **f**, **g**) Friedman test with posttest Dunn's multiple comparisons when significant results were obtained and (**e**) repeated measures ANOVA with posttest Tukey's multiple comparisons; (**h–j**) Paired t test and (**k**) Wilcoxon matched-pairs rank test; the p values are adjusted for multiple comparisons. Effect size measures ⁺⁺⁺high, ⁺⁺medium, ⁺small were determined by (**a–c**) $r_p$ – Pearson's correlation coefficient; (**d**, **f**, **g**) W – Kendall's W; (**e**) $\eta_p^2$ – partial eta-squared; (**h–j**) d – Cohen's d and (**k**) r – correlation coefficient r.

expression. Another possibility is that these HLA-DR:TCR interactions stabilize TLR4 expression at the T-cell plasma membrane. Further studies will be needed to dissect the mechanism by which HLA-DR regulates TLR4 expression on T cells. It is possible that HLA-DR:TCR interactions occur more frequently in the densely packed joint environment, where TLR4⁺ T cells are enriched. Suggesting the enticing possibility that HLA-DR mediated T:T-cell interactions might sensitize for joint microenvironment recognition and for a contextually driven shift of their pathological program. Interestingly, a recent paper has shown that T:T-cell interactions play a critical role in

driving IL-17 inflammatory responses in conditions of antigen scarcity[62].

TLR4 is a relatively promiscuous immune sensor that recognizes both microbial and endogenous ligands. This is in stark contrast with the TLR4⁺ Tfh-like cell population reported here; TLR4⁺ T cells were expanded in synovial fluid, and even though they were enriched for PD-1 they did not exhibit signs of either exhaustion or senescence, as illustrated by their highly proliferative status and increased ability to produce cytokines in response to stimulation. In addition to CXCR5, TLR4⁺ T cells also expressed the chemokine receptors CCR2 and CCR6

indicating preferential recruitment to inflamed tissues, which might account for their enrichment in the affected joints. Interestingly, TLR4 signaling has been reported to augment T-cell migration and invasiveness[63,64], opening the possibility that direct TLR4 engagement could propel T-cell invasiveness into the affected joints.

In mice models of autoimmune diseases, TLR4 signaling in CD4[+] T cells has been reported to function both as disease facilitator[24] and protector[23]. Nonetheless, a role for direct TLR4 engagement in T-cell cytokine profile and function had not been reported so far. Our data show that while TCR engagement favors the production of antibody-inducing cytokine IL-21, TLR4 engagement by either LPS or synovial fluid components ensues IL-17, IL-10, and TNF-α production, cytokine whose role in RA has been ascribed to promoting joint damage[8,12,42–45]. Even though IL-10 is often labeled as an anti-inflammatory cytokine, it is well established that IL-10 has both immunosuppressive and stimulatory effects, including cytotoxic activity against tumors[65]. In RA, IL-10 has been reported to drive inflammatory arthritis and joint destruction[66]. The existence of an antibody-independent pathogenic function for TLR4[+] T cells would explain why this population is also present in seronegative RA patients.

Curiously, although TLR4 engagement by LPS functions as a costimulatory signal boosting TCR signaling, TLR4 ligation by endogenous TLR4 ligands fuels TLR4[+] T-cell inflammatory program independently of cognate antigen recognition. Distinct ligands ensuing different TLR4 responses are likely owing to the fact that TLR4 has multiple binding sites[50]. In fact, TLR4 ligation by endogenous ligands TNC and fibronectin is not blocked by an LPS mimetic, which blocks TLR4 activation by competing with LPS for TLR4/MD-2 binding[19,67]. In addition, gene expression profiles induced by hyaluronan and TNC are significantly different from those induced by LPS[19,68,69]. Consistent with these previous reports, we observed that while TNC induced an IL-10 response, endogenous ligands present in synovial fluid favored IL-17 production. Even though, we cannot formally exclude that other components present in the synovial fluid might affect T-cell function, blocking TLR4 in the presence of synovial fluid completely abrogated (three out of five donors) or at the very least impaired (two out of five donors) IL-17 production. Thus, we can conclude that the production of IL-17 induced by the synovial fluid is specifically mediated by TLR4 on T cells. In view of our TNC results, it is likely that these TLR4 sponsored effects are mediated by the combined action of several endogenous TLR4 ligands present in the joints.

Importantly, ex vivo freshly analyzed synovial TLR4[+] T cells seemed to be skewed toward IL-17 production. When compared with in vitro stimulation with cell-depleted synovial fluid, synovial TLR4[+] T cells appear to be poised to produce more IL-17, less IL-10, and no TNF-α. These differences might be owing to the fact that to release cells from synovial fluid, it is necessary to degrade it enzymatically. Hyaluronidase digestion could give rise to additional TLR4 ligands that could be more adept at inducing IL-10 and TNF-α in vitro restimulation assays. In particular, different molecular weight hyaluronic acid fragments are known to elicit distinct inflammatory profiles[70]. It is possible that in vivo, IL-17 is the main cytokine induced by direct engagement of TLR4 on synovial T cells, where it might play a prominent role in mediating bone erosions and cartilage damage[71,72].

Our study employed a considerable RA patient cohort. Nonetheless, there are some limitations to our study. We could only obtain a relatively modest number of synovial fluid samples. This was due to the fact that we only used freshly obtained synovial fluid whose access to was seriously hindered during the COVID-19 pandemic. Another limitation was that most of the patients recruited presented controlled disease, which made it difficult to correlate the frequency of TLR4[+] T cells with disease activity. In addition, our functional assays were robustly identifying a causal relationship linking TLR4[+] T cells selective recognition of joint tissue environment to the type immune profile ensued. Further studies will be needed to address the impact of TLR4[+] T cells in joint damage.

Deciphering which CD4[+] T cells are relevant to the disease process and their interplay with the joint microenvironment is a critical hurdle to our understanding of RA. Here, we propose a mechanism by which the joint tissue microenvironment might reset on TLR4[+] T cells pathological function. Outside the joints, TLR4[+] Tfh-like cells will be activated predominantly through the TCR leading to the production of IL-21, which favors antibody production and will likely contribute to anti-CCP antibody titers. It is tempting to speculate that within the affected joints, T:T-cell interactions mediated through non-cognate HLA-DR:TCR-coupling supports TLR4 surface expression. In turn, direct sensing of joint damage patterns by TLR4[+] T cells reprograms them towards an IL-17 pathological program that drives and sustains cartilage damage and bone erosions. This two-prong mechanism could highlight several attractive therapeutic targets both at the systemic level and in the affected tissues. In addition, circulating TLR4[+] T cells in the blood could constitute a good biomarker to predict flares and possibly which patients are more likely to develop cartilage damage and joint erosions.

## Methods

**Human samples.** We have complied with all relevant ethical regulations and the Ethics Committee of NOVA Medical School (84/2019/CEFCM) and of Hospital Egas Moniz (20170700050) approved this study. Informed consent was obtained from RA patients that fulfilled ACR 2010 classification criteria and from HDs. RF status, C-reactive protein level, erythrocyte sedimentation rate, and medication usage were obtained by review of medical records. Anti-CCP antibody titers were determined at the time of blood draw using a commercial assay anti-CCP ELISA (IgG) from EUROIMMUN with a positive result defined as >5 RU/m. The number of swollen and/or tender joints was measured by the attending clinician on the day of sample acquisition. Treatments are categorized in: non-steroid anti-inflammatory (NSAID), corticosteroids, disease-modifying antirheumatic drugs (DMARDs), and biological DMARDs (dDMARDs). Blood was drawn by venipuncture into Lithium-Heparin containing cell preparation tubes (BD, Vacutainer). Synovial fluid was collected only when excess material was from patients undergoing diagnostic or therapeutic arthrocentesis. For cytokine experiments, patients under biological therapy or with combined DMARDs were not considered. One donor was excluded owing to % TLR4 expression <0.3%. Demographic and clinical data for all the patients enrolled in this study are listed in Supplementary Data 1 and for HDs are listed in Supplementary Table 1.

**Peripheral blood and synovial fluid cell isolation.** Blood samples and synovial fluid were processed within 4 h of collection and freshly analyzed. Peripheral blood and synovial mononuclear cells were isolated by density gradient centrifugation (Biocoll, Merck Millipore) or following enzymatic digestion with hyaluronidase (10 μL mL$^{-1}$; 30 min at 37 °C), respectively. Plasma and cell-depleted synovial fluid were frozen until further use.

**Antibodies and flow cytometry.** For flow cytometry analysis peripheral blood cells were stained with antibodies listed in Supplementary Data 2. For cell viability, Fixable Viability Dye (eBioscience) or Calcein Violet-AM (Biolegend) were used. When described, cells were cultured overnight with 10 μg mL$^{-1}$ of anti-HLA-DR antibody (L243). When mentioned FACS-purified CD14$^-$CD3$^+$CD4$^+$FSC$^{low}$S-SC$^{low}$ cells from RA patients or CD3$^+$CD4$^+$ cells from HDs were stimulated with anti-CD3 and anti-CD28 antibodies for 5/6 days and labeled for surface TLR4 expression. For proliferation assays, FACS-purified CD4$^+$ cells from HD were incubated in Cell Trace$^{TM}$ Violet dye according to the manufacturer's instructions. For intracellular staining, cells were treated with a Transcriptional Factor Fixation/Permeabilization kit (ebioscience). FACS acquisition was performed in a BD FACSCanto II instrument (BD Biosciences), BD FACSAria III (BD Biosciences), and further analyzed with FlowJo v10.7.1 software.

**Cell sorting and intracellular cytokine staining.** For flow cytometry cell sorting, cells were stained with anti-CD4 (RPA-T4) and anti-CD3 (SK7) antibodies (BioLegend); anti-CD4 (RPA-T4), anti-CD3 (SK7) and anti-HLA-DR (L243) or with anti-CD4 (SK3), anti-CD3 (UCHT1), and anti-CD14 (63D3) according to the

sorting strategy. Gating strategies are depicted in Supplementary Figs 1–3. Sorted populations cell purity was routinely >98%. For intracellular cytokines assays sorted $CD3^{high}CD4^{high}$, rested for at least 3 h, were stimulated with 5 μg mL$^{-1}$ of anti-CD3 (UCHT1, BioLegend) and 2 μg mL$^{-1}$ of anti-ICOS (C398.4 A, BioLegend), crosslinked with 5 μg/mL anti-mouse IgG1 (BioLegend) plus 10 μg mL$^{-1}$ anti-hamster IgG (Thermo Fisher Scientific) at 37°C in the presence of Brefeldin-A (Life Technologies) for 14 h. Cells were fixed in paraformaldehyde 1% (Sigma-Aldrich) and permeabilized with saponin (Carl Roth). Antibodies used are listed in Supplementary Data 2. When indicated 1.7 μg mL$^{-1}$ LPS (Sigma-Aldrich), TNC (Merck Millipore), or cell-depleted synovial fluid (SF) was added. For TLR4 blocking, CLI-095 (InvivoGen) was added at 10 μg mL$^{-1}$ 1 h before stimulation. Cell sorting was performed in a BD FACSAria III instrument (BD Biosciences).

**Imaging, image processing, and quantification**. FACS-purified $CD3^{high}CD4^{high}HLA-DR^+$ cells from RA patients or $CD3^+CD4^+$ cells from HDs were immediately plated onto poly-L-lysine–coated coverslips, fixed in 4% paraformaldehyde for 15 min at room temperature, incubated with blocking buffer (PBS BSA 1%), and immunostained as previously described[26,73]. Antibodies used for immunofluorescence staining are described in Supplementary Table 1. Confocal images were obtained using a Zeiss LSM 710 confocal microscope (Carl Zeiss) over a 63x objective. Z stack optical sections were acquired at 0.2 μm depth increments, and both green and red laser excitation were intercalated to minimize crosstalk between the acquired fluorescence channels. 3D image deconvolution was performed using Huygens Essential 19.10, and 2D images were generated from a maximum intensity projection over a 3D volume cut of 0.4-μm depth centered on the cell medium plane using Imaris. For quantification of cell size and roundness, confocal images were acquired at 2-μm increments in the z axis.

**Flow cytometry data analysis**. Flow cytometry data were analyzed using FlowJo and pluggins DownSample and FlowAI. The flow cytometry data were compensated at the time of acquisition with UltraComp eBeads (Thermo Fisher). As controls unstained and FMO conditions were included. The data collected in.fcs files were analyzed so that all abnormal events would be excluded by using FlowAI[74]. Then, by using the gating strategies mentioned in the figures, dead cells and doublets were excluded. Whenever mentioned ΔMFI was calculated by subtracting the FMO from median fluorescence intensity (MFI) for any given fluorophore being analyzed. t-SNE maps were generated by pooling patients and the final $n$ is described in the legend. Every heatmap represents differential marker expression between TLR4$^+$ cells (dashed gate) and remaining CD4$^+$ T-cell populations. To maintain the consistency of the events from each condition and also to reduce the number of events fed into t-SNE algorithm, DownSample was used and files were concatenated in a way that all conditions/donors could be represented in the same plot.

**Statistics and reproducibility**. Statistical analysis was performed by using GraphPad Prism v9.00 and IBM SPSS Statistics version 26 software's. First, we tested the normality of the data by using D'Agostino & Pearson normality test ($n > 6$) or Shapiro–Wilk normality test ($n \leq 6$), by checking skewness and kurtosis values and visual inspection of data. Then, if the samples followed a normal distribution, we chose the appropriate parametric test; otherwise, the non-parametric counterpart was chosen. We also verified the assumptions that are required for each test. Then, by using ROUT method ($Q = 1\%$) that is provided in GraphPad we checked for the existence of outliers. All tests were performed with and without outliers to verify their effect on $p$ value and the results pointed in the same direction (significant and non-significant values). We report here the results that include the outliers. All statistical tests performed were two-tailed.

Overall, a $p$ value ≤ 0.05 ($\alpha$) was considered statistically significant. All the analyses developed a 95% confidence interval. The $p$ values were calculated using the true distribution (exact $p$ values). Results were considered significant at *$p \leq 0.05$, **$p \leq 0.01$, ***$p \leq 0.001$, ****$p \leq 0.0001$. For multiple comparisons, adjusted $p$ values were used.

In two groups comparison: for paired data, Paired $t$ test, Ratio-paired $t$ test or Wilcoxon matched-pairs signed-rank test was used; for unpaired data, Mann–Whitney test was used. For multiple groups comparison: for paired data, Repeated measures one-way analysis of variance (ANOVA) with posttest Turkey's multiple comparisons or Friedman test with posttest Dunn's multiple comparisons was used; for unpaired data Kruskal–Wallis test with posttest Dunn's multiple comparisons were used as indicated. For correlations, Pearson or Spearman was used as described. The choice of each test was dependent on the underlying distribution and is indicated in the legend of the figures.

Data are presented as mean ± standard deviation (SD), for parametric statistical tests and median ± interquartile range (IQR) for non-parametric statistical tests. The number of biological replicates ($n$) is specified in the legend of the figure. As a measure of the magnitude of the difference, the effect size was calculated as described[75,76]:

-For Paired $t$ test/ratio-paired $t$ test: Cohen's $d$ ($d$) is small if <0.3; medium if ≥0.3 and <0.8 or large if ≥0.8;

-For Wilcoxon matched-pairs signed-rank test/Mann–Whitney test: correlation coefficient $r$ ($r$) is small if <0.3; medium if ≥0.3 and <0.5 or large if ≥0.5;

-For repeated measures ANOVA: partial eta-squared ($\eta_p^2$) is small if ≥0.01 and <0.06; medium if ≥0.06 and <0.14 or large if ≥0.14;

- For Friedman test: Kendall's $W$ ($W$) is small if ≥0.1 and <0.3; medium if ≥0.3 and <0.5 or large if ≥0.5;

-For Kruskal–Wallis test: eta square ($\eta^2$) is small if ≥0.01 and <0.06; medium if ≥0.06 and <0.14 or large if ≥0.14;

-For Pearson correlation: Pearson's correlation coefficient ($r_p$): small if <0.3; medium if ≥0.3 and <0.5 or large if ≥0.5;

-For Spearman correlation: Spearman's correlation coefficient ($r_s$): small if <0.3; medium if ≥0.3 and <0.5 or large if ≥0.5.

The effect sizes values are reported in the figures and are labeled as $^+$ for small, $^{++}$ for medium, and $^{+++}$ for large effect sizes according to these values.

A table with a statistic summary per figure can be found in Supplementary Data 3.

**Reporting summary**. Further information on research design is available in the Nature Research Reporting Summary linked to this article.

## Data availability
The data sets generated during and/or analyzed during the current study are available as Supplementary data files.

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

## Acknowledgements

We thank Cláudia Andrade for technical support and Juliana Gonçalves for testing samples for SARS-CoV-2 exposure. We are extremely grateful to all the participants of the study and to the whole rheumatology department at Hospital Egas Moniz that made this study possible. This work was supported by Fundação para a Ciência e Tecnologia (FCT) PTDC/MEC-REU/29520/2017, by iNOVA4Health UID/Multi/04462 and by Portuguese Society for Rheumatology (SPR) grants to H.S. H.S. is supported by FCT through IF/01722/2013 and CEECIND/01049/2020, DAS and RCT were supported by FCT through PD/BD/137409/2018 and UID/Multi/04462, respectively.

## Author contributions

D.A.S., R.G., R.C.T. designed and performed experiments and D.A.S. analyzed the data. R.T. collected and analyzed clinical data. A.N., I.S., S.F., M.C., N.P.G., R.T., M.J.M. recruited patients and provided blood and synovial recruited patients, provided blood and SF samples. A.B.S., C.L., M.M., M.H.L., M.J.G., P.A., S.M., T.C., W.C. recruited patients and provided blood samples. F.P.S. and A.F.M. advised, recruited patients, provided blood samples, analyzed, and discussed clinical data. J.C.B. advised, analyzed, and interpreted clinical data. H.S. conceived and supervised the project, designed and performed experiments, analyzed the data, and wrote the manuscript. All authors discussed the results and commented on the manuscript.

## Competing interests

The authors declare no competing interests.
