## [Transparent Peer Review File · Communications Biology]

Reviewers' comments:

Reviewer #2 (Remarks to the Author):

In this study, Amaral-Silva et al identifies and characterize TLR4 expressing CD4 T cells in patients with rheumatoid arthritis. The authors show that TLR4 expression by CD4 T cells is associated with an activated phenotype, with increased size and high expression of HLA-DR, CD38 and proliferation marker KI-67. In addition, they find that TLR4 expressing CD4 T cells have a Tfh like phenotype and express several chemokine receptors, which supports their enrichment in the synovium of rheumatoid arthritis patients. Importantly, they provide evidence that CD4 T cells can be activated by synovial fluid from rheumatoid arthritis patients in a TLR4 dependent manner, supporting their claim that this population of CD4 T cells can play an important role in disease pathogenesis.

Overall, it's a well performed and interesting study that provides evidence for a new pathway of T cell activation during rheumatoid arthritis. However, there are some concerns that needs to be addressed especially regarding the gating strategy for identifying TLR4 expressing CD4 T cells. My comments are listed below.

Specific comments

1. In figure 1 the authors provide a detailed characterization of TLR4 expressing CD4 T cells in patients with rheumatoid arthritis. Although, this characterization is thorough I do have some concerns regarding the flow cytometry gating strategy. It seems like all the TLR4 expressing CD4 T cells fall within what would normally be considered a "doublet" gate. Although the authors do provide an explanation for why these cells are included in the text and the confocal imaging supports the expression of TLR4, I do find that there are still concerns. Did the authors include antibodies for other immune cells, for example monocytes and B cells? The gating strategy is not stated in the figure legend and it therefore seems like the authors only used live dead, CD3 and CD4 before evaluating TLR4 expression. Since the classical doublet gate is included it is possible that monocytes (which express high levels of TLR4) have been included in the gating. To address this concern the authors should either state if other markers were used in their analysis or perform a new staining which includes for example a monocyte marker to exclude that other TLR4 expressing cells have been included in the gating.

2. The authors evaluate TLR4 expressing CD4 T cells in patients with rheumatoid arthritis. What is the frequency of TLR4 expressing CD4 T cells in healthy individual? Since it seems like all HLA-DR positive CD4 T cells express TLR4 in patients with rheumatoid arthritis, do HLA-DR positive T cells in healthy individuals also express TLR4 to a similar extent? To provide additional evidence for a potential role of TLR4 expressing CD4 T cells in the pathogenesis of rheumatoid arthritis the authors should include information regarding the frequency of this cells in the peripheral blood of healthy individuals.

3. The authors should include isotype or FMO staining controls for several of the flow cytometry gating panels. This is especially important for panels where gates are different for TLR4+ and TLR4- cells, for example most of the panels in figure 4.

4. The majority of HLA-DR expressing CD4 T cells express TLR4 and the authors do provide a functional connection between HLA-DR and TLR4 expression in CD4 T cells, which indicates an interesting functional role for HLA-DR. Since blocking HLA-DR downregulates TLR4 expression, one would expect that also the sensitivity to TLR4 stimulation would be affected. To strengthen their claim of a functional role for HLA-DR in T cells and the connection with TLR4, the authors could evaluate if blocking HLA-DR would affect the response towards TLR4 stimulation in CD4 T cells.

5. The story would benefit from further characterization of the role of TLR4 in CD4 T cells. For example, if naïve CD4 T cells are activated will they upregulate TLR4? Do certain cytokine promote TLR4 expression? Do TLR4 stimulation promote proliferation of CD4 T cells? This is information that

would further support a functional role for TLR4 in T cells.

6. In figure 7 the authors provide important evidence that CD4 T cells can be activated by TLR4 ligands that present in the synovial fluid of rheumatoid arthritis patients. This is important since it shows a functional role for TLR4 in CD4 T cells. How many patients were evaluated in panels H to O? It seems like 2 in some panels and 3 in some. Due to the rather larger variability between the samples, especially in panels H-K, the authors should increase the number of patients evaluated.

Reviewer #3 (Remarks to the Author):

This paper describes activated T cells in RA synovial fluid (and blood) that express class II MHC and TLR4. Experiments are performed to assess their role in disease, but much remains uncertain. I have a few specific questions.

1- Is there anything unique about TLR4 expression on T cells in RA compared to other inflammatory diseases? Or compared to normal T cells activated by polyclonal stimuli?

2- Are these cells a biomarker for disease activity? Longitudinal studies of a few patients would help to answer this.

3- Are synovial fluid TLR4+ T cells representative of similar cells in synovial tissue? In part this could be answered by immunostaining of the tissue.

4- Do these cells also make interferon gamma, either with IL-21 or with IL-17? Intra-cellular staining for 2 cytokines simultaneously would be useful.

5- What is the effect of endogenous TLR4 ligands that are found in the RA joint on the cytokine production pattern? LPS would not normally be expected in the RA joint.

6- It is not convincing to assume that effect of anti-Class II antibody is due to blocking homotypic adhesion.

Such antibody can generate a signal to the cell on its own. Also, there are reports that MHC, when citrullinated can function as a signaling ligand for other types of receptors (Holoshitz et al). The authors could start by showing that homotypic adhesion of these cells is blocked by anti-Class II. I would guess that such adhesion is more likely dependent on LFA-1/ICAM, CD2/CD58, CD6/CD166 interactions etc.

Lisbon, August 4th 2021

Manuscript ID: COMMSBIO-21-0335-T

Title: "Direct tissue sensing reprograms TLR4⁺ Tfh-like cells inflammatory profile in the joints of rheumatoid arthritis patients"

Dear Reviewers,

We are grateful for the constructive revisions that we received. Before elaborating our revision plan, we consulted with our handling editor to gauge the scope of the revision, particularly, in the context of restricted access to patients' samples due to the ongoing restrictions imposed by COVID-19 pandemic (in person hospital appointments are still strongly discouraged). Within these restrictions, we strove to fully satisfy the maximum possible reviewer's comments. This included recruitment of additional RA patients and healthy donors, performing several new experiments which resulted in one completely new figure, two new supplementary figures, one new demographic table, and one figure that was completely reformulated (added *n* and added panels). We have made a point-by-point answer to the reviewers detailing the efforts made to address the reviewers' concerns. Moreover, we improved our statistical analysis and included a size effects measures to every figure panel.

Thank you for your time.

Sincerely,

Helena Soares, Pharm.D., Ph.D.

Group Leader of Human Immunobiology and Pathogenesis Lab
Invited Assistant Professor of Immunology
Chronic Disease Research Centre (CEDOC)
NOVA Medical School, NOVA University of Lisbon

Amaral-Silva et al. COMMBIO-21-0335-T
point-by-point answer to reviewers

Reviewer #2 (Remarks to the Author):

In this study, Amaral-Silva et al identifies and characterize TLR4 expressing CD4 T cells in patients with rheumatoid arthritis. The authors show that TLR4 expression by CD4 T cells is associated with an activated phenotype, with increased size and high expression of HLA-DR, CD38 and proliferation marker KI-67. In addition, they find that TLR4 expressing CD4 T cells have a Tfh like phenotype and express several chemokine receptors, which supports their enrichment in the synovium of rheumatoid arthritis patients. Importantly, they provide evidence that CD4 T cells can be activated by synovial fluid from rheumatoid arthritis patients in a TLR4 dependent manner, supporting their claim that this population of CD4 T cells can play an important role in disease pathogenesis.

Overall, it's a well performed and interesting study that provides evidence for a new pathway of T cell activation during rheumatoid arthritis. However, there are some concerns that needs to be addressed especially regarding the gating strategy for identifying TLR4 expressing CD4 T cells. My comments are listed below.

Specific comments

Reviewer 2.1 In figure 1 the authors provide a detailed characterization of TLR4 expressing CD4 T cells in patients with rheumatoid arthritis. Although, this characterization is thorough I do have some concerns regarding the flow cytometry gating strategy. It seems like all the TLR4 expressing CD4 T cells fall within what would normally be considered a "doublet" gate. Although the authors do provide an explanation for why these cells are included in the text and the confocal imaging supports the expression of TLR4, I do find that there are still concerns. Did the authors include antibodies for other immune cells, for example monocytes and B cells? The gating strategy is not stated in the figure legend and it therefore seems like the authors only used live dead, CD3 and CD4 before evaluating TLR4 expression. Since the classical doublet gate is included it is possible that monocytes (which express high levels of TLR4) have been included in the gating. To address this concern the authors should either state if other markers were used in their analysis or perform a new staining which includes for example a monocyte marker to exclude that other TLR4 expressing cells have been included in the gating.

Authors 2.1 *We thank the reviewer for this comment. At the reviewer's suggestion, we added the monocyte marker and TLR4 co-receptor, CD14, to our staining panel. We found that, in fact, CD4⁺TLR4⁺ T cells co-express CD14 (Fig. 2a). We reasoned that CD14 labelling could have two origins: it could be the result of T cell:monocytes conjugates, OR, a subset of CD3⁺CD4⁺ T cells, from RA patients, could gain the expression of both CD14 and TLR4. Our microscopy results (Fig 1k) strongly suggested the latter possibility. Nonetheless, to address the reviewers' concerns, we dwelled on this matter further and designed a new experiment in which we FACS-purified CD14*

CD3⁺CD4⁺ T cells using as gating strategy a restricted lymphocyte gate (low FSC and SSC values) to exclude doublets and then excluded CD14⁺ cells (**Fig S2a**). To formally show that these highly purified CD14⁻CD3⁺CD4⁺ T cells could indeed acquire CD14/TLR4 expression we either left them unstimulated or stimulated them for 5 days with α CD3 plus α CD28 antibodies. While unstimulated cells only marginally acquired CD14 expression after 5 days in culture, TCR stimulation led to the novo expression of CD14 by a subset of CD4⁺ T cells from RA donors (**Fig 2b**), whose frequency (2-4%) is similar to the one we had found for TLR4⁺CD4⁺ T cells in RA patients' circulation (**Fig 1d**). Of note, and similarly to what we had found for ex vivo TLR4 expression (**Fig 1d**), CD14 expression by CD3⁺CD4⁺ T cells was restricted to the cells with higher SSC-A values. Moreover, CD14 expression by previously CD14⁻CD3⁺CD4⁺ T cells was dependent on TCR stimulation, suggesting that CD14/TLR4 expression acquisition by CD4⁺ T cells is under the control of TCR and thus subsequent to T cell activation state. We also showed that this was the case for CD4⁺ T cells from healthy donors (see answer 2.2) albeit to a lesser extent, likely attributable to the differences in overall activation state verified between RA patients and healthy donors. Together with our microscopy data, where TLR4 colocalized with TCR at the cellular membrane, these data strongly suggest that the hyperactivation state of RA patients leads to the differentiation of a pathogenic T cell subset characterized by the expression of myeloid markers CD14 and TLR4. This set of experiments afforded interesting data and an insight of how TLR4 expression is acquired by CD4⁺ T cells. However, it intrinsically denoted a limitation, as we were impaired to address the reviewer's suggestion to use CD14 as a sorting exclusion marker, since in addition to excluding monocytes it would also exclude our T cell subset of interest.

Reviewer 2.2. The authors evaluate TLR4 expressing CD4 T cells in patients with rheumatoid arthritis. What is the frequency of TLR4 expressing CD4 T cells in healthy individual? Since it seems like all HLA-DR positive CD4 T cells express TLR4 in patients with rheumatoid arthritis, do HLA-DR positive T cells in healthy individuals also express TLR4 to a similar extent? To provide additional evidence for a potential role of TLR4 expressing CD4 T cells in the pathogenesis of rheumatoid arthritis the authors should include information regarding the frequency of this cells in the peripheral blood of healthy individuals.

Authors 2.2. We thank the reviewer for the suggestion and in fact, we had already looked at TLR4 expression in healthy patients and found it to be undetectable ex vivo (**Fig. 2 c, d**). We went a step further and checked whether TLR4 expression could be induced in healthy donors (HD), similarly to what we had determined for RA CD4⁺ T cells (**Fig 2b**) upon 5-6 days of TCR stimulation. We FACS purified CD3⁺CD4⁺ T cells from HD (**Fig S2c**) and stimulated them with α CD3 plus α CD28 antibodies. Interestingly, CD3⁺CD4⁺ T cells from HD gained TLR4 surface expression (**Fig 2d**), albeit at inferior levels than the one observed in CD4⁺ T cells from RA patients. This was well illustrated by the fact that while TLR4 expression in RA CD4⁺ T cells could be readily detectable by flow cytometry, TLR4 expression in HD CD4⁺ T cells could only be assessed by confocal microscopy, as TLR4 levels in HD CD4⁺ T cells were below the detection threshold for flow cytometry. In addition, we complemented this phenotypic characterization

with functional assays and determined that TLR4 expression on HD CD4⁺ T cells is functional, as supplementation of LPS increased HD CD3⁺CD4⁺ T cell size (Fig 2e), survival (Fig 2f), and proliferative capacity (Fig 2 g, h). Thus, HD CD3⁺CD4⁺ T cells under basal conditions do not express TLR4. However, upon TCR stimulation HD CD3⁺CD4⁺ T cells gain TLR4 at their cellular surface, albeit to a much lesser extent than the one found in circulation in RA patients. Lastly, TLR4 expression by HD CD3⁺CD4⁺ T cells is functional, and it mediated increase in T cell size, survival and proliferation.

Reviewer 2.3. The authors should include isotype or FMO staining controls for several of the flow cytometry gating panels. This is especially important for panels where gates are different for TLR4⁺ and TLR4⁻ cells, for example most of the panels in figure 4.

Authors 2.3. *We fully agree with the reviewer. We had based our gating strategy on FMO values but had not showed them. We have now added FMO staining controls for all the panels in figure 4 (Fig S4).*

Reviewer 2.4. The majority of HLA-DR expressing CD4 T cells express TLR4 and the authors do provide a functional connection between HLA-DR and TLR4 expression in CD4 T cells, which indicates an interesting functional role for HLA-DR. Since blocking HLA-DR downregulates TLR4 expression, one would expect that also the sensitivity to TLR4 stimulation would be affected. To strengthen their claim of a functional role for HLA-DR in T cells and the connection with TLR4, the authors could evaluate if blocking HLA-DR would affect the response towards TLR4 stimulation in CD4 T cells.

Authors 2.4. *We thank the reviewer for the suggestion and we have done the experiment. As showed in Fig 4e by blocking HLA-DR, we completely loose TLR4 expression. As we lose the sole molecular marker that defines the population of interest, it is impossible to determine its cytokine profile. In an extra attempt, we even tried to stain for intracellular TLR4, reasoning that HLA-DR blocking might lead to TLR4 internalization rather than regulate expression. However, HLA-DR blocking also impaired the detection of intracellular TLR4, suggesting that in fact HLA-DR is needed and regulates TLR4 expression, rather than its surface expression. Due to the low frequency of TLR4⁺ T cells in the donors recruited for the paper revision, we did not have enough cell numbers to perform this experiment with sorted TLR4⁺ T cells.*

Reviewer 2.5. The story would benefit from further characterization of the role of TLR4 in CD4 T cells. For example, if naïve CD4 T cells are activated will they upregulate TLR4? Do certain cytokine promote TLR4 expression? Do TLR4 stimulation promote proliferation of CD4 T cells? This is information that would further support a functional role for TLR4 in T cells.

Authors 2.5. *We thank the reviewer and we do believe that the manuscript is further enriched by these data. In figure 2, we now show that TLR4 expression can be induced on resting CD4⁺ T cells, even from healthy donors, upon TCR and CD28 stimulation. Once upregulated TLR4 increases T cell size, survival and proliferation (Fig 2 c-h), in addition of leading to the reprogramming of TLR4⁺ T cell inflammatory profile (Figs 7, 8).*

Reviewer 2.6. In figure 7 the authors provide important evidence that CD4 T cells can be activated by TLR4 ligands that present in the synovial fluid of rheumatoid arthritis patients. This is important since it shows a functional role for TLR4 in CD4 T cells. How many patients were evaluated in panels H to O? It seems like 2 in some panels and 3 in some. Due to the rather larger variability between the samples, especially in panels H-K, the authors should increase the number of patients evaluated.

Authors 2.6. *We thank the reviewer for the feedback. We would like to state that former Fig 7 is currently Fig 8. Even though it was really challenging to recruit more patients in the midst of the COVID-19 pandemic (hospital medical appointments are still strongly discouraged and still occur by videoconference) we were capable of increasing our n by 2-3 fold, the n in panels m-p is now of 6 and in panels q-t it varies between 5 and 6.*

Reviewer #3 (Remarks to the Author):

This paper describes activated T cells in RA synovial fluid (and blood) that express class II MHC and TLR4. Experiments are performed to assess their role in disease, but much remains uncertain. I have a few specific questions.

Reviewer 3.1- Is there anything unique about TLR4 expression on T cells in RA compared to other inflammatory diseases? Or compared to normal T cells activated by polyclonal stimuli?

Authors 3.1. *We are currently finishing another paper addressing TLR4⁺ T cells presence and role in another inflammatory disease. As presented in Figure 2 and discussed in answers 2.2 and 2.5 to reviewer #2, T cells from healthy donors upregulate TLR4 upon CD3 and CD28 engagement, albeit to a lesser extent than the one observed in RA patients. We functionally addressed the role of TLR4 engagement in CD4⁺ T cells from healthy donors and found that it increases cell size, cell survival and their proliferative capability. Nonetheless, we want to underline that polyclonal stimulation of CD4⁺ T cells from healthy donors does not lead to the high levels of TLR4 expression observed in CD4⁺ T cells from RA patients, or from other human inflammatory diseases that we are currently studying. It is tempting to propose that under physiological conditions there might be a regulatory cap for TLR4 expression by CD4⁺ T cells, or that in order to achieve higher TLR4 expression levels CD4⁺ T cells might require additional signals (inflammatory cytokines?) in addition to CD3 and CD28 stimulation. This difference in expression levels between healthy donors and RA patients might have important pathophysiological properties, for instance, while in CD4⁺ T cells from HD TLR4 functions as a coreceptor, enhancing TCR stimuli, in RA patients, TLR4 functions independently of TCR engagement, as depicted in Fig 7 c, g and Fig 8 i, m. Another very important difference is that the lesser expression levels of TLR4 in CD4⁺ T cells from healthy donors might preempt TLR4 inflammatory reprogramming observed in RA patients (Fig 7, 8).*

Reviewer 3.2- Are these cells a biomarker for disease activity? Longitudinal studies of a few patients would help to answer this.

Authors 3.2. *We agree with the reviewer that it would be really interesting to check whether TLR4 on T cells could function as biomarker for disease progression and/or treatment efficacy. Nonetheless, we feel that such a long study is out of the scope of this manuscript and it will be better suited as a follow-up study. In addition, it would be really challenging to implement while the restrictions of COVID-19 are still being enforced. Currently, following directives from the Portuguese government, all hospital consultations are being held by videoconference, which makes it extremely hard patients' recruitment and follow-up.*

Reviewer 3.3- Are synovial fluid TLR4+ T cells representative of similar cells in synovial tissue? In part this could be answered by immunostaining of the tissue.

Authors 3.3. *We agree that checking for TLR4⁺ T cells presence in the synovium tissue would enrich the manuscript. Unfortunately, due to the COVID-19 restrictions for these last 18 months, which lead to the suspension of all non-essential/urgent hospital activities in Portuguese hospitals, arthroscopies were suspended unless deemed essential/urgent. As such, we were not able to gain access to synovial tissue from RA patients and were unable to address the reviewer's request. We feel confident that all the other experiments that we performed at the reviewers' bequest have made the paper more solid and enhanced its scope.*

Reviewer 3.4- Do these cells also make interferon gamma, either with IL-21 or with IL-17? Intra-cellular staining for 2 cytokines simultaneously would be useful.

Authors 3.4. *We thank the reviewer for the suggestion. Early on, in this project we had tried to stain for IFN- γ but were not able to detect it. Now, following the reviewer suggestion we have stained for IFN- γ upon TLR4⁺ T cell with the TLR4 ligand tenascin-C, and once again we did not detect IFN- γ production (**Fig 8l**). As for double producers, we found that TLR4⁺ T cells either produce IL-17 or IL-21. When stimulated with LPS we could detect some IL-17⁺TNF- α ⁺ double positive producers.*

Reviewer 3.5- What is the effect of endogenous TLR4 ligands that are found in the RA joint on the cytokine production pattern? LPS would not normally be expected in the RA joint.

Authors 3.5. *This is a very important point that we had, at least partially, addressed by showing that stimulation with cell depleted synovial fluid there originated IL-17 production and that this cytokine production was TLR4 dependent since it was blocked by the TLR4 specific inhibitor CLI-095 (**Fig 8m**). We believe that we have now made these results stronger by increasing the n of the experiment (**Fig 8m**). Furthermore, and consubstantiating the reviewer's point, we have added a stimulatory condition to **Figure 8** with the known TLR4 endogenous ligand Tenascin-C, which has been implicated in*

RA. We show that like cell-free synovial fluid, Tenascin-C induces cytokine production by TLR4⁺ T.

Reviewer 3.6- It is not convincing to assume that effect of anti-Class II antibody is due to blocking homotypic adhesion. Such antibody can generate a signal to the cell on its own. Also, there are reports that MHC, when citrullinated can function as a signaling ligand for other types of receptors (Holoshitz et al). The authors could start by showing that homotypic adhesion of these cells is blocked by anti-Class II. I would guess that such adhesion is more likely dependent on LFA-1/ICAM, CD2/CD58, CD6/CD166 interactions etc.

Authors 3.6. *We thank the reviewer for the comment and we have now rewritten the manuscript removing such assumption. We now discuss Holoshitz paper as one possible explanation for HLA-DR mediated mechanism to regulate TLR4 expression. Nonetheless, our new findings present in **Figure 2**, in which TCR stimulation increases CD14/TLR4 expression in CD4⁺ T cells from both RA patients and HD reinforces our view that HLA-DR might be regulating TLR4 expression through T:T cell non-cognate interactions. Curiously, a recent paper by Boccasavia et al showed that antigen presentation between T cells drives Th17 polarization, which is reminiscent of the IL-17 reprogramming of TLR4⁺ T cells that we describe here.*

Figure 2. TCR and CD28 stimulation leads to CD14 and TLR4 upregulation in RA patients and healthy donors. **a** Representative dot plot and cumulative graph of frequency of CD14 expression in CD3⁺CD4⁺ T cells of peripheral blood of RA patients (n = 6). **b** Representative dot plot and cumulative graph of CD14 expression by FACS purified CD14⁺CD3⁺CD4⁺SSC⁺FSC⁺T cells from RA patients after 1 and 5 days with and without αCD3 and αCD28 stimulation (n = 4). **c-d** Confocal microscopy of FACS-purified CD3⁺CD4⁺SSC⁺FSC⁺T cells from healthy donors (HD) that were either left unstimulated (unstim) or were stimulated with αTCR and αCD28 for 5 days in the presence and absence of LPS. **c** Cells were surface labeled for CD45 and TLR4 and analyzed by 3D confocal microscopy. Bar, 5 μm. **d** Cumulative graph of 3D TLR4 fluorescence (n = 143 cells from 4 different HD). **e** Representative histogram and cumulative plot of FSC-A MFI of CD4⁺ T from HD stimulated for 5 days with αTCR and αCD28 in the presence or absence of LPS (n = 29 HD, 53 independent experiments). **f** Representative dot plots and cumulative graph of the frequency of viable cells CD4⁺ T from HD stimulated for 5 days with αTCR and αCD28 in the presence or absence of LPS and labelled with cell viability dye (n = 39). **g** Representative histogram and cumulative plot of cell trace MFI of CD4⁺ T from HD stimulated for 5 days with αTCR and αCD28 in the presence or absence of LPS (n = 5). **h** Representative dot plots and cumulative graph of proliferative CD4⁺ T from HD stimulated for 5 days with αTCR and αCD28 in the presence or absence of LPS (n = 5). Data is presented as mean ± SD, for parametric statistical tests, or median ± IQR, for non-parametric statistical tests. Sample normality distribution was tested by using D'Agostino & Pearson normality test (n > 6) or Shapiro-Wilk normality test (n ≤ 6). P-values ****p ≤ 0.0001, ***p ≤ 0.001, **p ≤ 0.01, *p ≤ 0.05 were determined by (b) Ratio-paired t test; (d) Kruskal-Wallis test with posttest Dunn's multiple comparisons; the p-values are adjusted for multiple comparisons; (e, g, h) Wilcoxon matched-pairs rank test and (f) Paired t-test. Effect size measures: ***high, **medium, *small were determined by (b, f) d - Cohen's d, (d) η² - eta-squared and (e, g, h) r - correlation coefficient r.

Figure 8. Direct recognition of TLR4 ligands present in synovial fluid drives IL-17 production, independently of antigen recognition. a-c Correlation between synovial fluid tenascin-C levels and a DMARDs duration ($n = 6$), b frequency of circulating (PB) TLR4⁺ T cells ($n = 7$), and c frequency of synovial fluid (SF) TLR4⁺ T cells ($n = 7$). d-g FACS-purified CD3^{hi}CD4^{hi} T cells from peripheral blood were cultured for 18 hours in the presence of medium (Med) or TLR4 signaling inhibitor (CLI-095). Frequency and Δ MFI of IL-17 (d, e) and IL-10 (f, g) production by TLR4⁺ T cells ($n = 10$). h-l FACS-purified CD3^{hi}CD4^{hi} T cells from peripheral blood were cultured for 18 hours in the presence of medium (Med) or with the TLR4 endogenous ligand tenascin-C (TNC). Frequency of h IL-17, i IL-10, j TNF- α , k IL-21 and l IFN- γ production by TLR4⁺ T cells ($n = 6$). m-p FACS-purified CD3^{hi}CD4^{hi} T cells from peripheral blood were cultured for 18 hours in the presence of medium (Med), synovial fluid (SF) or TLR4 signaling inhibitor (CLI-095). Frequency of m IL-17, n IL-10, o TNF- α and p IL-21 production by TLR4⁺ T cells ($n = 5$). q-t Ex vivo production of q IL-17 ($n = 6$), r IL-10 ($n = 5$), s TNF- α ($n = 5$ RA) and t IL-21 ($n = 5$) by TLR4⁺ T cells in freshly obtained peripheral blood (PB) and synovial fluid (SF) donor paired samples. Δ MFI was calculated by subtracting the fluorescence intensity minus one (FMO) from median fluorescence intensity (MFI) for each given marker. FMOs were calculated independently for blood and synovial fluid FACS analysis. Data is presented as mean \pm SD, for parametric statistical tests, or median \pm IQR, for non-parametric statistical tests. Sample normality distribution was tested by using D'Agostino & Pearson normality test ($n > 6$) or Shapiro-Wilk normality test ($n \leq 6$). P-values **** $p \leq 0.0001$, *** $p \leq 0.001$, ** $p \leq 0.01$, * $p \leq 0.05$ were determined by (a, b, c) Pearson Correlation; (d, e, f, g, i, k, q, r, s) Paired t-test; (h, j, l, t) Wilcoxon matched-pairs rank test; (m, o, p) Friedman test with posttest Dunn's multiple comparisons when significant results were obtained and (n) Repeated measures ANOVA with posttest Tukey's multiple comparisons; the p-values are adjusted for multiple comparisons. Effect size measures ****high, **medium, *small were determined by (a, b, c) r_p - Pearson's correlation coefficient; (d, e, f, g, i, k, q, r, s) d - Cohen's d ; (h, j, l, t) r - correlation coefficient r ; (m, o, p) W - Kendall's W and (n) η^2_p - partial eta-squared.

Supplementary Figure 2. FACS-purification strategy and sorted cell population purity of RA patient and healthy donor. **a** Flow cytometric sorting strategy for the purification of CD14⁺ CD3⁺ CD4⁺ SSC^{low} FSC^{low} T cells from RA patients. **b** Purity of sorted CD14⁺ CD3⁺ CD4⁺ SSC^{low} FSC^{low} T cells from RA patients. **c** Flow cytometric sorting strategy for the purification of CD3^{high} CD4^{high} FSC^{low} SSC^{low} T cells from healthy donor. **d** Purity of sorted CD3^{high} CD4^{high} FSC^{low} SSC^{low} T cells from healthy donor.

Supplementary Figure 4. Fluorescence minus one (FMO) conditions for gating Tfh markers in TLR4⁻ and TLR4⁺ T cell populations. **a** Representative FMO dot plot for CXCR5 in TLR4⁻ (grey) versus TLR4⁺ (red) T cells, with gating illustrating the individual positivity cut-off for both cell populations. **b** Representative FMO dot plot for PD-1 in TLR4⁻ (grey) versus TLR4⁺ (red) T cells, with gating illustrating the individual positivity cut-off for both cell populations. **c** Representative FMO dot plot for ICOS in TLR4⁻ (grey) versus TLR4⁺ (red) T cells, with gating illustrating the individual positivity cut-off for both cell populations.

REVIEWERS' COMMENTS:

Reviewer #2 (Remarks to the Author):

The authors have responded to all my comments.

Reviewer #3 (Remarks to the Author):

The paper is improved, but would benefit from synovial tissue immunohistology

Lisbon, 19th August 2021

RE: Revision of the manuscript COMMSBIO-21-0335-A “Direct tissue sensing reprograms TLR4⁺ Tfh-like cells inflammatory profile in the joints of rheumatoid arthritis patients”

Dear Reviewers,

Once again, we would like to acknowledge your constructive comments. We were pleased that Reviewer 2 was satisfied by our revisions, and we will reinforce what we had already stated in the previous revision regarding the single remaining concern expressed by reviewer 3. Please find our detailed response in below.

Thank you for your time.

Sincerely,

Helena Soares, Pharm.D., Ph.D.
Group Leader of Human Immunobiology and Pathogenesis Lab
Invited Assistant Professor of Immunology
Chronic Disease Research Centre (CEDOC)
NOVA Medical School, NOVA University of Lisbon

Reviewer #3: The paper is improved, but would benefit from synovial tissue immunohistology

Authors: We fully agree with the reviewer. Providing synovial tissue immunohistology would have enable us to strengthen the message of the paper. Unfortunately, this one only one (and, in global perspective, a very minor one at that) of the limitations imposed by the COVID-19 pandemic. It will be something that we will look forward pursuing in the future and thank the reviewer for highlighting this.